# Diversity vs. Recognizability: Human-like generalization in one-shot generative models

**Victor Boutin**[1,2], **Lakshya Singhal**[2], **Xavier Thomas**[2] and **Thomas Serre**[1,2]
[1] Artificial and Natural Intelligence Toulouse Institute, Université de Toulouse, France
[2] Carney Institute for Brain Science, Dpt. of Cognitive Linguistic & Psychological Sciences
Brown University, Providence, RI 02912
{victor_boutin, thomas_serre}@brown.edu

## Abstract

Robust generalization to new concepts has long remained a distinctive feature of human intelligence. However, recent progress in deep generative models has now led to neural architectures capable of synthesizing novel instances of unknown visual concepts from a single training example. Yet, a more precise comparison between these models and humans is not possible because existing performance metrics for generative models (i.e., FID, IS, likelihood) are not appropriate for the one-shot generation scenario. Here, we propose a new framework to evaluate one-shot generative models along two axes: sample *recognizability* vs. *diversity* (i.e., intra-class variability). Using this framework, we perform a systematic evaluation of representative one-shot generative models on the Omniglot handwritten dataset. We first show that GAN-like and VAE-like models fall on opposite ends of the diversity-recognizability space. Extensive analyses of the effect of key model parameters further revealed that spatial attention and context integration have a linear contribution to the diversity-recognizability trade-off. In contrast, disentanglement transports the model along a parabolic curve that could be used to maximize recognizability. Using the diversity-recognizability framework, we were able to identify models and parameters that closely approximate human data.

## 1 Introduction

Our ability to learn and generalize from a limited number of samples is a hallmark of human cognition. In language, scientists have long highlighted how little training data children need in comparison to the richness and complexity of the language they learn so efficiently to master [12, 40]. Similarly, children and adults alike are able to learn novel object categories from as little as a single training example [17, 8]. From a computational point of view, such feats are remarkable because they suggest that learners must be relying on inductive biases to overcome such challenges [33] – from an ability to detect suspicious coincidences or 'non-accidental' features [43, 52] to exploiting the principle of compositionality [32, 33].

While a common criticism of modern AI approaches is their reliance on large training datasets, progress in one-shot categorization has been significant. One-shot categorization involves predicting an image category based on a unique training sample per class. Multiple algorithms have been proposed including meta-learning algorithms [18, 46, 14, 37] or metric-learning algorithms [47, 31, 48] that are now starting to approach human accuracy. Perhaps a less studied problem is the one-shot generation problem – aimed at creating new variations of a prototypical shape seen only once. Since the seminal work of Lake et al. [32] who introduced the Bayesian Program Learning algorithm, only a handful of promising one-shot generative algorithms have been proposed [42, 15, 1] (see section 2.2 for a more exhaustive description of prior work).

36th Conference on Neural Information Processing Systems (NeurIPS 2022).

Why have so few algorithms for one-shot image generation vs. image categorization been proposed? We argue that one of the main reasons for this lack of progress is the absence of an adequate evaluation metric. As of today, one-shot generative models are evaluated using methods initially developed for models producing samples that belong to the training categories and trained on large datasets. Those metrics include the likelihood, the FID (Frechet Inception Distance), or the IS (Inception Score). In the one-shot image generation scenario in which training images are scarce and the generated samples represent new visual concepts, the likelihood, the FID, and the IS are biased [3, 13, 38] (see 2.1 for more details). These limitations urge us to look for new metrics tailored for one-shot image generation.

Recent psychophysics work [52] has characterized humans' ability for one-shot generation along two main axes: samples *diversity* (i.e., intra-class variability) and samples *recognizability* (i.e., how easy or hard they are to classify). According to this framework, ideal generalization corresponds to a combination of high recognizability and high diversity. As illustrated in Fig. 1, an ideal model should be able to generate samples that span the entire space within the decision boundary of a classifier (Box 1). In comparison, the model of Box 2 has learned to make identical copies of the prototype (i.e., low diversity but high accuracy). Such a model has failed to generalize the visual concept exemplified by the prototype. Similarly, if the model's samples are so diverse that they cannot be recognized accurately as shown in the Box 3 of Fig. 1, then the generated samples won't look like the prototype.

Here, we borrow from this work and adapt it to create the first framework to evaluate and compare humans and one-shot generative models. Using this framework, we systematically evaluate an array of representative one-shot generative models on the Omniglot dataset [32]. We show that GAN-like and VAE-like one-shot generative models fall on opposite ends of the diversity-recognizability space: GAN-like models fall on the high recognizability — low diversity end of the space while VAE-like models fall on the low recognizability — high diversity end of the space. We further study some key model parameters that

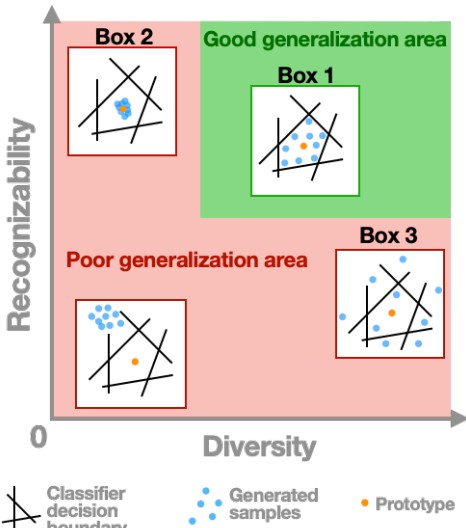

Figure 1: **The diversity vs. recognizability framework.** The best possible samples for good generalization (green area) are those that match the intra-class variations (i.e., remain within decision boundaries; Box 1). Bad samples associated with poor generalization (red area) include strategies that involve exact copies of the prototype (Box 2) and samples that exceed the intra-class variability (Box 3).

modulate spatial attention, context integration, and disentanglement. Our results suggest that spatial attention and context have an (almost) linear effect on the diversity vs. recognizability trade-off. In contrast, varying the disentanglement moves the models on a parabolic curve that could be used to maximize the recognizability. Last but not least, we have leveraged the diversity vs. recognizability space to identify models and parameters that best approximate the human data on the Omniglot handwritten dataset.

## 2 Related work

### 2.1 Metrics to evaluate generative models and their limitations for one-shot generation tasks

Different types of generative models are typically evaluated using different metrics. On the one hand, likelihood-based algorithms (e.g., VAE [29], PixelCNN [39], GLOW [30], etc.) are evaluated using their own objective function applied on a testing set. Likelihood provides a direct estimate of the KL divergence between the data points and the model's samples. On the other hand, implicit generative models such as Generative Adversarial Networks (GANs) [20] for which the loss function cannot be used, are typically evaluated using other scores such as the Inception Score (IS) [45] or Frechet Inception Distance (FID) [22]. IS and FID are heuristic measures used to aggregate both the sample quality and diversity in one single score. The IS scores a sample quality according to the

confidence with which an Inception v3 Net [49] assigns the correct class label to it. The FID score is the Wasserstein-2 distance between two Gaussian distributions: one fitted on the features of the data distribution and the other on the features of the model distribution (the features are also extracted from an Inception v3 Network).

All these metrics are problematic for the one-shot generation scenario for 2 main reasons that are intrinsically related to the task: the low number of samples per class, and the dissimilarity between training and testing visual concepts. IS and FID rely on statistical distances (either KL divergence for IS or Wasserstein-2 distance for FID) that require a high number of data points $N$ to produce an unbiased estimator of the distance. Even when used in the traditional settings (i.e., $N = 50000$), it has been demonstrated that both scores are biased [13]. This is to be compared with the $N = 20$ samples typically available in popular few-shot learning datasets such as Omniglot [32]. Another problem caused by the limited number of samples per class in the training set is the overfitting of the Inception Net used to extract the features to compute the IS and FID [7]. To illustrate this phenomenon we have conducted a small control experiment in which we have trained a standard classifier to recognize images from the Omniglot datasets (see S17). In this experiment, we have used 18 samples per classes for training and 2 samples per classes for testing. In Fig. S21a, we observe an increase of the testing loss while the training loss is decreasing. This is a clear sign of overfitting. Note that this overfitting is not happening when the standard classifier is replaced by a one-shot (or few-shot) classifier. This control experiment show that standard classifier are not adapted to extract relevant features in the low-data regime. Consequently, IS and FID are not suitable in the low-data regime.

The second limitation of these metrics appears because the training and the testing samples are too dissimilar. Likelihood scores are known to yield higher scores for out-of-domain data compared to in-domain data [38]. Therefore, the evaluation of the novel visual concepts generated by one-shot generative models will be biased toward higher scores. In addition, both FID and IS rely on distance between features extracted by an Inception Net which comes with no guarantee that it will produce meaningful features for novel categories. For example, class misalignment has been reported when the Inception Net was trained on ImageNet and tested on CIFAR10 [3]. Because of all the aforementioned limitations, it is pretty clear that new procedures are needed to evaluate the performance of few-shot generative algorithms.

## 2.2 One-shot generative models

One can distinguish between two broad classes of one-shot generative models: structured models and statistical models [16]. Structured models have strong inductive biases and rigid parametric assumptions based on a priori knowledge such as for example a given hierarchy of features, a known grammar or program [44]. A prominent example of a structured model includes the very first algorithm for one-shot image generation, the Bayesian Program Learning (BPL) model [32]. Statistical models learn visual concepts by learning statistical regularities between observed patterns [42, 15, 19]. Here, we focus on representative architectures of one-shot generative statistical models, which we summarize below.

- VAE with Spatial Transformer Network (**VAE-STN**) [42]. The VAE-STN is a sequential and conditional Variational Auto-Encoder (VAE) constructing images iteratively. The VAE-STN algorithm uses a recurrent neural network (i.e., an LSTM) to encode the sequence of local patches extracted by an attentional module. A key ingredient of the VAE-STN is an attention module composed of a Spatial Transformer Network (STN) [26] to learn to shift attention to different locations of the input image. The STN is a trainable module to learn all possible affine transformations (i.e., translation, scaling, rotation, shearing) of an input image (see S8 for samples and details of the **VAE-STN**).

- Neural statistician (**VAE-NS**) [15]: The Neural Statistician is an extension of the conditional VAE model including contextual information. Therefore, in addition to learning an approximate inference network over latent variables for every image in the set (as done in a VAE), the approximate inference is also implemented over another latent variable, called the context variable, that is specific to the considered visual concept. The context inference network is fed with a small set of images representing variations of a given visual concept. The **VAE-NS** has been extended to include attention and hierarchical factorization of the generative process [19] (see S9 for samples and details of the **VAE-NS**).

- Data-Augmentation GAN (DA-GAN) [1]: Data-Augmentation GAN is a generative adversarial network conditioned on a prototype image. The DA-GAN generator is fed with a concatenation of a vector drawn from a normal distribution and a compressed representation of the prototype. The discriminator is trained to differentiate images produced by the generator from images of the dataset, while the generator has to fool the discriminator. We have trained 2 different DA-GAN, one is based on the U-Net architecture (**DA-GAN-UN**) and the other one on the ResNet architecture (**DA-GAN-RN**) (see S10 for samples and details of the **DA-GAN-UN** and S11 for samples and details of the **DA-GAN-RN**).

All these models are generative models conditioned by an image prototype extracted from the training or the test set. The way we have selected the prototypes is detailed in Eq. 1. To the best of our knowledge, these models offer a representative set of one-shot generative models. We have reproduced all these models (sometimes with our own implementation when it was not available online). Our code could be found at `https://github.com/serre-lab/diversity_vs_recognizability`.

## 3 The diversity vs. accuracy framework

Let $\{x_i^j\}$ be a dataset composed of K concepts (i.e., classes) with N samples each ($i \in [\![1, N]\!]$ and $j \in [\![1, K]\!]$). The framework we propose aims at evaluating the performance of a generative model $p_\theta$, parameterized by $\theta$, that produces new images $v_i^j$ based on a single sample (or prototype) of a concept given to the generator $\tilde{x}^j$ (i.e., $v_i^j \sim p_\theta(\cdot|\tilde{x}^j)$). For each concept $j$, we define a prototype as the sample closest to the center of mass for the concept $j$:

$$\tilde{x}^j = x_{i^*}^j \quad \text{s.t.} \quad i^* = \operatorname*{argmin}_i \left\| f(x_i^j) - \frac{1}{N} \sum_{i=1}^N f(x_i^j) \right\|_2 \tag{1}$$

In Eq. 1, $f$ denotes a function that projects the input image from the pixel space to a feature space. We will detail the feature extractor $f$ shortly. Note that this definition of a prototype is not unique (one could also select the prototype randomly within individual classes $j$), nevertheless this selection mechanism is a guarantee that the selected sample will be representative of the concept.

**Dataset.** In this article, we use the Omniglot dataset [32] with a weak generalization split [42]. Omniglot is composed of binary images representing $1,623$ classes of handwritten letters and symbols (extracted from $50$ different alphabets) with just $20$ samples per class. We have downsampled the original dataset to be $50 \times 50$ pixels. The weak generalization split consists of a training set composed of all available symbols minus 3 symbols per alphabet which are left aside for the test set. It is said to be *weak* because all the alphabets were shown during the training process (albeit not all symbols in those alphabets). As the Omniglot dataset is hand-written by humans, we consider that these samples reflect a human generative process and we refer to this later as the **human** model.

**Diversity.** In the proposed framework, *diversity* refers to the intra-class variability of the samples produced by a generative model $p_\theta(\cdot|\tilde{x}^j)$. For a given prototype $\tilde{x}^j$, we compute the diversity as the standard deviation of the generated samples in the feature space $f$:

$$\sigma_{p_\theta}^j = \sqrt{\frac{1}{N-1} \sum_{i=1}^N \left( f(v_i^j) - \frac{1}{N} \sum_{i=1}^N f(v_i^j) \right)^2} \quad \text{s.t.} \quad v_i^j \sim p_\theta(\cdot|\tilde{x}^j) \tag{2}$$

We use the Bessel-corrected standard deviation to keep a good estimate of the data dispersion despite the relatively small number of samples (e.g., $N = 20$ for the Omniglot dataset used here). To verify that this diversity measure is robust to the specific choice of the feature extractor $f$, we explored two different settings: features learned with class supervision by a Prototypical Net [47] and features learned with self-supervision by a SimCLR network [10]. In both cases, we extracted the features from the first fully-connected layer following the last convolutional layer. The Prototypical Net was optimized so that images that belong to the same category share similar latent representations as measured by the $\ell_2$-norm. Similarly, SimCLR leverages a contrastive loss to define a latent representation such that a sample is more similar to its augmented version than to other image samples. In SimCLR, this similarity is computed with cosine similarity. These two approaches represent two ends of a continuum of methods to learn suitable representational spaces without the

need to explicitly learn to classify images and are thus more suitable for few-shot learning tasks [35]. For the sake of comparison, we have used the exact same network architecture for both feature extractors (see sections S1 and S2 for more details on Prototypical Net and SimCLR, respectively).

We computed the samples diversity for all 150 categories of the Omniglot test set (i.e., $v_i^j = x_i^j$ in this experiment) using both the supervised and unsupervised settings. We found a high linear correlation ($\rho = 0.86$, p-value $< 10^{-5}$) and a high rank-order Spearman correlation ($\rho = 0.85$, p-value $< 10^{-5}$) between the two settings (see section S3.1). Hence, the two feature extraction methods produce comparable diversity measures and henceforth, we will report results using the unsupervised setting.

As an additional control, we have also verified that the SimCLR metric is robust to changes to the augmentation method used (see section S3.3) and to the specific choice of the dispersion metric (see section S3.2 for more details on this comparison). We have compared the feature space of the Prototypical Net and SimCLR using a t-SNE analysis (see section S3.4). We observed a strong clustering of samples belonging to the same category for both networks. It suggests that the augmentation methods used by the SimCLR contrastive loss are sufficient to disentangle the class information.

Fig. 2 shows the 10 concepts from the Omniglot test set with the lowest and highest samples diversity, respectively (for more diversity-ranked concepts with unsupervised or supervised setting, see sections S4 and S5, respectively). One can see that the proposed diversity metric is qualitatively similar to human judgment. Concepts with low diversity are composed of very few relatively basic strokes (e.g., lines, dots, etc) with little room for any kind of "creativity" in the generation process while more diverse concepts are composed of more numerous and more complex stroke combinations with many more opportunities for creativity.

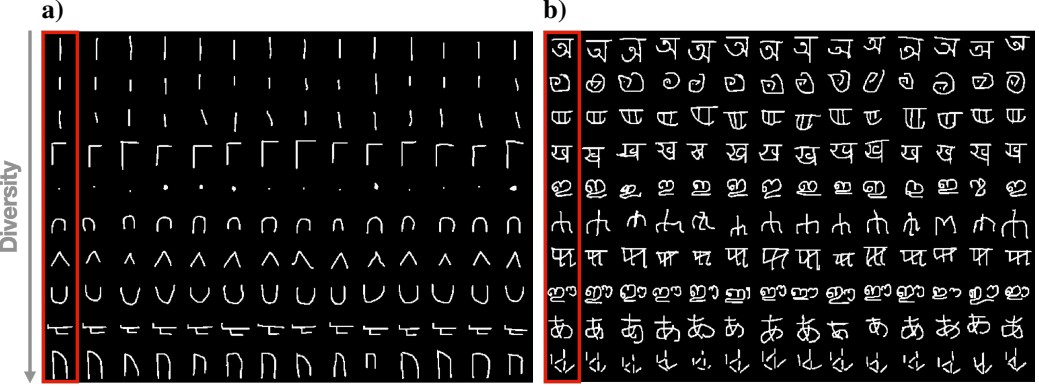

Figure 2: Samples from the top 10 Omniglot concepts (test set) associated with the lowest (**a**) vs. highest diversity (**b**). The different concepts are ranked vertically from less diverse to more diverse. Prototypes for individual concepts are shown within a red box next to actual class samples.

**Recognizability.** We evaluate the recognizability of the samples produced by the one-shot generative models by leveraging one-shot classification models. As demonstrated in S17, it is also possible to use a few-shot classifier to evaluate the recognizability. We prefer one-shot classifier to match the settings proposed in [32]. In order to make sure our classification accuracy measure is robust to the choice of the model, we test different models which belong to the two main approaches used in machine learning for one-shot classification: metric learning and meta-learning [35]. We selected the Prototypical Net [47] as a representative metric-learning approach and the Model-Agnostic Meta-Learning (MAML) [18] model as a representative meta-learning approach. Both models were trained and tested in a 1-shot 20-ways setting (see section S6 for more details on the MAML architecture and training details). We report a high Pearson (negative) correlation between the logits produced by Prototypical Net and MAML ($\rho = -0.60$, p-value $< 10^{-5}$) as well as a strong Spearman rank-order correlation between the classification accuracy of both networks ($\rho = 0.62$, p-value $< 10^{-5}$). See section S7 for more details about this control experiment. Hence, our recognizability metric is robust to the choice of the one-shot classification model (even when those models are leveraging different approaches) and henceforth, we will report results using the Prototypical Net model.

# 4 Results

## 4.1 GAN-like vs. VAE-like models

For all algorithms listed in section 2.2 we have explored different hyper-parameters (see section 4.2 for more details), leading to various models represented in the diversity vs. recognizability plot in Fig. 3a. In this figure, we have reduced each model to a single point by averaging the diversity and recognizability over all classes of the Omniglot testing set. The black star corresponds to the **human** model, and colored data points are computed based on the samples generated by the **VAE-NS**, **VAE-STN**, **DA-GAN-UN** and the **DA-GAN-RN**. The base architectures for all algorithms (highlighted with bigger points in Fig. 3a) have a comparable number of parameters (≈6-7 M, see S8, S9 and S10 for more details on the base architectures).

We observe that the GAN-like models (i.e., **DA-GAN-UN** and **DA-GAN-RN**) tend to be located at the upper left side of the graph while VAE-like models (i.e., **VAE-NS** and **VAE-STN**) spread on the right side of the graph. Therefore, the GAN-like models produce very recognizable samples that are highly similar to each other (high recognizability and low diversity). In contrast, VAE-like models generate more diverse but less recognizable samples. The samples in Fig. 3b illustrate this observation. The difference between GAN and VAE-like samples could be explained by their loss functions [36]. The GANs' adversarial loss tends to drop some of the modes of the training distribution. In general, the distribution learned by GANs put excessive mass on the more likely modes but discards secondary modes [2]. This phenomenon leads to sharp and recognizable generations at the cost of reduced samples diversity. On the other hand, VAEs (and likelihood-based models in general) are suffering from over-generalization: they cover all the modes of the training distribution and put mass in spurious regions [4]. We refer the reader to Fig. 4 of Lucas et al. [36] for an illustration of mode dropping in GANs and over-generalization in VAEs. Our diversity vs. recognizability plot in Fig. 3a shows that this phenomenon is holding even when the testing distribution is different from the training distribution as in the case of the one-shot generation scenario.

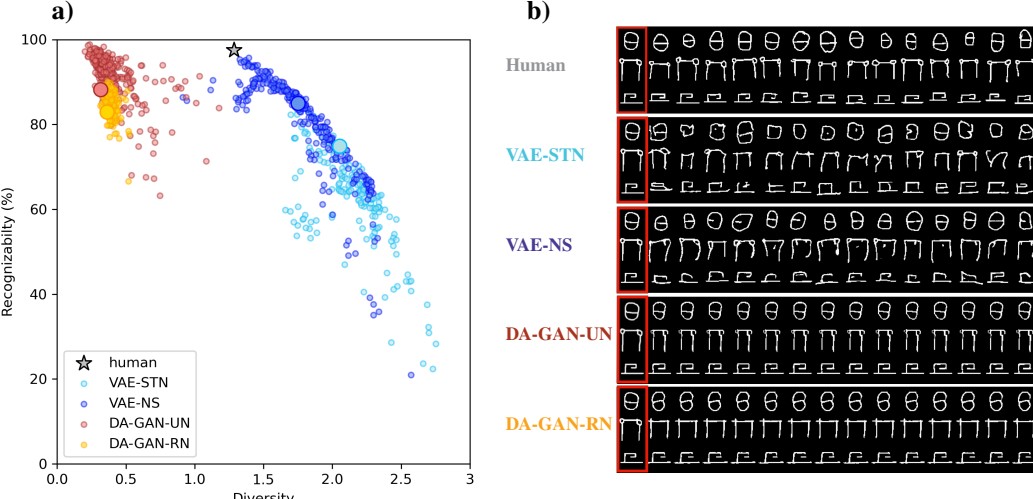

Figure 3: (**a**) Diversity vs. recognizability plot for all tested models (colored data points) and human (black star). Each data point corresponds to the mean diversity and recognizability over all classes of the Omniglot test set. Bigger circles correspond to the base architecture of each model that all have a comparable number of parameters ($\approx 6 - 7M$). The **human** data-point is computed based on the testing samples of the Omniglot dataset. (**b**) Samples produced by the different models in their base architectures (corresponding to the bigger circles in Fig. 3a). Prototypes for individual concepts are shown within a red box next to actual class samples.

## 4.2 VAE-NS vs. VAE-STN

In this section we compare some key hyper-parameters of the **VAE-NS** and **VAE-STN**. The core idea of the **VAE-NS** is to integrate context information to the sample generation process. During

training, the context is composed of several samples that all represent variations of the same visual concept. Those samples are passed in a separate encoder to extract a context statistics (denoted $c$ in S9) used to condition the generative process. During the testing phase, the **VAE-NS** infers the context statistics using a single image (i.e., the prototype). We evaluate the effect of the context on the position of the **VAE-NS** models on the diversity-recognizability space by varying the number of samples used to compute the context statistics during the training phase (from 2 to 20 samples). Importantly, varying the number of context samples do not change the number of parameters of the network. For all tested runs, we observe a monotonic decrease of the samples diversity (see Fig. S13a) and a monotonic increase of the samples recognizability (see Fig. S13b) when the number of context samples is increased. In the diversity-recognizability space, the resulting curve is monotonically transporting models from the lower-right side to the upper-left side of the plot (see Fig. 4a, dark blue curve). The effect of the number of context samples is large: the diversity is almost divided by 2 (from 2.4 to 1.2) and the classification accuracy is increased by 80% (from 53% to 96%). This result suggests that increasing the number of context samples for a given visual concept helps the generative model to identify the properties and the features that are crucial for good recognition, but hurts the diversity of the generated samples.

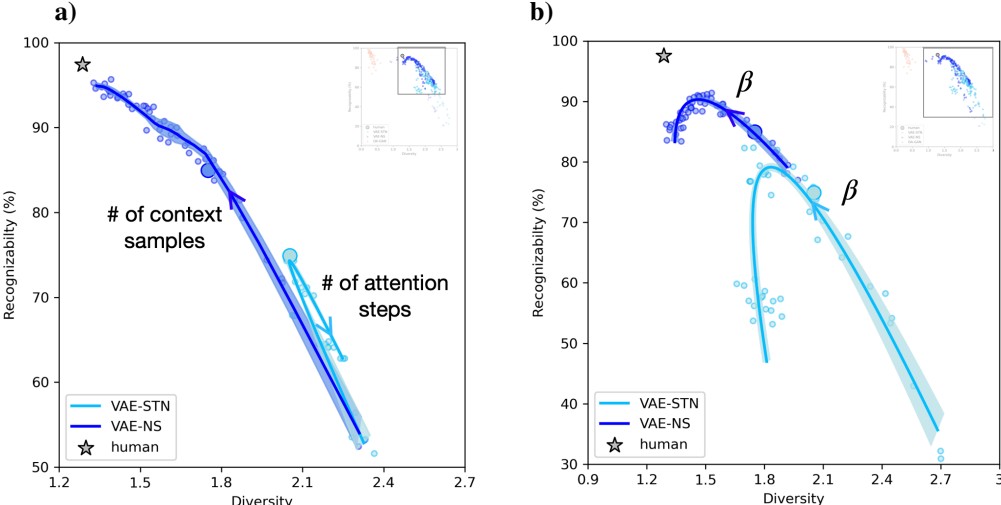

Figure 4: (**a**) Effect of the number of context samples of the **VAE-NS** and attentional steps of the **VAE-STN**. Each data point represents a model with a different number of context samples for the **VAE-NS** (ranging from 2 to 20) or a different number of attentional steps for the **VAE-STN** (ranging from 20 to 90). The base architectures, highlighted with a bigger circle, correspond to 10 context samples and 60 attentional steps for the **VAE-NS** and **VAE-STN** respectively. (**b**) Effect of $\beta$. The base architectures correspond to a $\beta = 1$. In all curves, solid lines represent the mean of the parametric curves over 3 different runs. Shaded areas are computing using the standard deviation over 3 different runs. Arrows show the direction in which the tested variables (context samples, attention steps or $\beta$) are increased.

In contrast to the **VAE-NS**, the **VAE-STN** uses spatial attention to sequentially attend to sub-parts of the image to decompose it into simpler elements. These sub-parts are then easier to encode and synthesize. In the **VAE-STN**, one can vary the number of attentional steps (i.e., the number of attended locations) to modulate spatial attention. Importantly, varying the number of attentional steps does not change the number of parameters. We have varied the number of attentional steps from 20 to 90. The relationship between the number of attentional steps, the samples diversity, and the samples recognizability is non-monotonic. We have used a parametric curve fitting method (i.e., the least curve fitting method from [21]) to parameterize the curve while maintaining the order of the data point (see S13 for more details on the fitting procedure). We report a convex parabolic relationship between the number of attentional steps and the samples diversity (see Fig. S14a). This curve is minimal at 60 steps. We observe a concave parabolic relationship between the number of attentional steps and the recognizability of samples. This curve is maximal at 60 attentional steps (see Fig. S14b). In Fig. 4b we have plotted the parametric fit illustrating the position of the **VAE-STN** models in the

diversity-recognizability space when one increases the number of attentional steps (the light blue curve). This curve follows a quasi-linear trend with a sharp turn-around (at 60 attentional steps). The effect of the number of attentional steps on the diversity-recognizability is limited compared to the effect of the number of context samples.

Both **VAE-NS** and **VAE-STN** are trained to maximize the Evidence Lower Bound (ELBO), it is then possible to tune the weight of the prior in the loss function. One can operate such a modulation by changing the $\beta$ coefficient in the ELBO loss function [24]. We refer the reader to S15 for more mathematical details about the ELBO. A high $\beta$ value enforces the latent variable to be closer to a normal distribution and increases the information bottleneck in the latent space. Increasing $\beta$ is known to force the disentanglement of the generative factors [9]. We observe a monotonic decreasing relationship between the value of $\beta$ and the samples diversity for both the **VAE-STN** and the **VAE-NS** (see Fig. S15a and Fig. S16a, respectively). We report a concave parabolic relationship between $\beta$ and the samples recognizability. We use the least curve fitting method to find the optimal parabolic curves [21]. This curve is maximal at $\beta = 2.5$ for the **VAE-STN** and at $\beta = 3$ for the **VAE-NS** (see Fig. S15b and Fig. S16b, respectively). The overall effect of $\beta$ on the position of the VAE-like models on the diversity-recognizability space is relatively similar for both the **VAE-STN** and the **VAE-NS** and follows a clear parabolic trend. These curves demonstrate that one could modulate the value of $\beta$ to maximize the recognizability. In general, we observe that the variable controlling the context-size in the **VAE-NS** is the one having the biggest impact on the diversity-recognizability space.

We have also varied the architecture of the **DA-GAN-UN**, **DA-GAN-RN**, **VAE-NS**, and **VAE-STN** by changing the size of the latent space. We did not find any common trend between the size of the latent variable, the diversity, and the recognizability (see S16 for more details). We observe that the **DA-GAN-UN** tends to produce slightly more recognizable but less diverse samples than the **DA-GAN-RN** while both architectures have the same number of parameters. It suggests that the extra skip connections included in the U-Net architecture, in between the encoder and the decoder of the **DA-GAN-UN**, allow to trade diversity for recognizability.

## 4.3 Comparison with humans

We now compare the tested models with the **human** data in the diversity-recognizability space. To perform such a comparison, we first normalize all the model's diversity and recognizability (including humans) using the z-score such that both axes are scaled and centered similarly. Then, for all models, we compute the $\ell_2$-distance between models and humans in the diversity-recognizability space. We remind that the **human** data point is computed using the samples of the Omniglot test set. Distances to humans as well as their distributions are reported for all models in Fig. 5a. The median of **VAE-NS** models is closer to humans, followed by **DA-GAN-UN**, **DA-GAN-RN** and **VAE-STN** (medians are indicated in Fig. 5a with horizontal bars). The **VAE-NS** model showing the smallest distance is almost at the human level (see dark blue square Fig. 5a). It has a context size of 20 samples (the highest possible context size), and a $\beta = 2.5$. The **VAE-STN** model that best approximates human has a $\beta = 2.25$ and 60 attentional steps (see light blue square in Fig. 5a).

So far, we have reduced all models to single points by averaging the diversity and recognizability values over all classes. We now study distances to humans for individual classes and for the **VAE-NS** and the **VAE-STN** models showing the shortest distance to humans (indicated by blue squares in Fig. 5a). In Fig. 5b, we report distances to human for these 2 models and for 16 visual concepts. The visual concepts 1 to 8 and 9 to 16 are selected so that they minimize the distance to humans with the **VAE-STN** and the **VAE-NS**, respectively. We observe that these visual concepts are different for the **VAE-NS** and the **VAE-STN** model. Therefore, both models are well approximating human data for some visual concepts but not for others. Interestingly, we qualitatively observe that the visual concepts 1 to 8 look simpler (i.e., made with fewer strokes) than the visual concepts 9 to 16. It suggests that the spatial attention mechanism used by **VAE-STN** provides a better human approximation for simple visual concepts, while the context integration method leveraged by the **VAE-NS** is more relevant to mimic human data on more complex visual concepts.

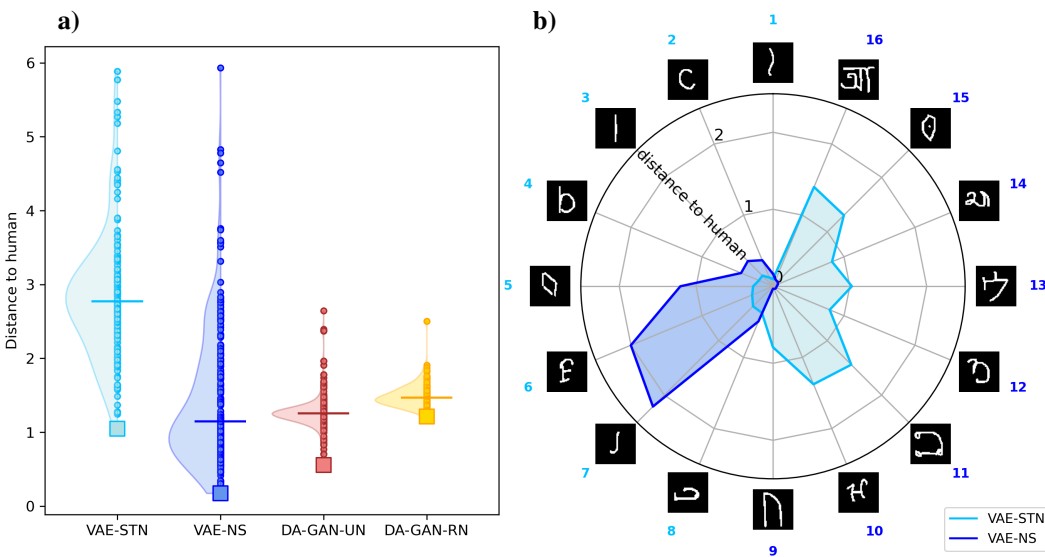

Figure 5: **(a)** Distribution of average distance to humans for the **VAE-NS**, **VAE-STN**, **DA-GAN-UN** and **DA-GAN-RN**. Each data point corresponds to the mean diversity and accuracy over all classes of the Omniglot test set. Squares correspond to the model showing the smallest distance to humans. The distance to humans is calculated with a $\ell_2$-norm on the diversity-recognizability space after z-score normalization. The horizontal line denotes the median of the model's distribution. **(b)** Distance to humans on 16 different visual concepts for the **VAE-NS** and **VAE-STN** models that best approximate human data (i.e., indicated by a square in Fig. 5a). The visual concepts 1 to 8 are selected to minimize the **VAE-STN** distance to humans, and visual concepts 9 to 16 minimize the **VAE-NS** distance to humans. Images surrounding the radar plot are the prototypes of the visual concepts.

## 5 Discussion

In this article, we have described a novel framework for comparing computational models with human participants on the one-shot generation task. The framework measures the diversity and the recognizability of the produced samples using metrics compatible with the one-shot scenario. To the best of our knowledge, this is the first and only framework specifically tailored to evaluate and compare one-shot image generation models.

Among all tested algorithms, the **VAE-NS** is the best human approximator on Omniglot (see Fig. 5a). It suggests that the context integration mechanism of the **VAE-NS** is an important component to reach human-like generalization. Interestingly, motor learning experiments have demonstrated that human generalization performances are also strongly related to contextual information [50]. Interestingly, [51] have demonstrated that a bayesian observer tends to overestimate the intra-class variance when only a few context samples are accessible. Our results are in-line with this finding: Fig. 4a shows an high diversity when the number of context samples is low while the diversity is decreasing when more context samples are available. It suggests that the **VAE-NS** is acting as a bayesian observer: it overestimates intra-class variance when the context is scarse.

In addition, we demonstrate that one can tune $\beta$ so that the model becomes closer to human data (see Fig. 4b). This is consistent with a prior computational neuroscience study that has shown that disentangled VAEs (with $\beta > 1$) provide a good model of face-tuned neurons in the inferotemporal cortex [23]. Our comparison between the **VAE-NS** and the **VAE-STN** suggests that a model which uses a spatial attention mechanism better fits human data for simple visual concepts. In contrast, the context integration mechanism of the **VAE-NS** appears to be a better human approximator for more complex visual concepts. One could thus try to combine both mechanisms towards improving the similarity with human data independent of the complexity of the visual concept. We have also found that GAN-like models (**DA-GAN-RN** and **DA-GAN-UN**) better account for human recognizability but do not approximate well the diversity of the human samples. In contrast, VAE-like models

(**VAE-NS** and **VAE-STN**) better account for human diversity. An interesting approach would be to leverage a hybrid architecture (such as the VAE-GAN [34]) to try to better match human data.

Other candidate ingredients include the ability to harness compositionality [32] or the recurrent processes thought to be crucial for human generalization [54]. Compositionality could be introduced in one-shot generative algorithms by quantizing the latent space (as in the VQVAE [53]). As a result, each coordinate of the latent variable represents an address in a codebook, and the role of the prior is then to combine simpler concepts to generate more complex samples. One promising way to include recurrent processing into generative models is through the predictive coding framework [41]. Predictive Coding suggests that each processing step is part of an inference scheme that minimizes the prediction error [6]. Previous work has demonstrated that such networks are more robust and exhibit improved generalization abilities [5, 11]. All these ingredients could be tested and compared against human abilities using the diversity/recognizability framework we have proposed in this paper.

In the current version of the Omniglot dataset, the intra-class variability does not reflect the human level of creativity. It is mainly due to the experimental protocol in which one asks human participants to copy a given visual concept. The Omniglot dataset could be enriched with more diverse samples, by explicitly asking human participants to be as creative as possible. Other drawing databases with more complex symbols such as *Quick Draw!* [27] could also be considered to strengthen the comparison with humans.

By decomposing the performance of the one-shot generation task along the recognizability vs. diversity axes we wanted to shed light on the relationship between generalization and creativity (quantified by the samples diversity in our framework). We hope one can make use of our framework to validate key hypotheses about human generalization abilities so that we can better understand the brain. We argue that the best way to reach human-like generalization abilities is to unleash the algorithms' creativity.

## Acknowledgement

This work was funded by ANITI (Artificial and Natural Intelligence Toulouse Institute) and the French National Research Agency, under the grant agreement number : ANR-19-PI3A-0004. Additional funding to TS was provided by ONR (N00014-19-1-2029) and NSF (IIS-1912280 and EAR-1925481). Computing hardware supported by NIH Office of the Director grant S10OD025181 via the Center for Computation and Visualization (CCV). We thanks Roland W. Fleming and his team for the insightful feedback and discussion about the diversity vs. recognizability framework.

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
