# Supplementary Information

## S1   More details on Prototypical Net

### Architecture

Table S1 describes the architecture of the Propotypical Net [47] we are using in this article. We use the Pytorch convention to describe the layers of the network.

Table S1: Description of the Prototypical Net Architecture

| Network | Layer | # params |
|---|---|---|
| ConvBlock(In$_c$, Out$_c$) | Conv2d(In$_c$, Out$_c$, 3, padding=1) | In$_c$ × Out$_c$ × 3 × 3 + Out$_c$ |
| | BatchNorm2d(Out$_c$) | 2 x Out$_c$ |
| | ReLU | - |
| | MaxPool2d(2, 2) | - |
| Prototypical Net | ConvBlock(1, 64) | 0.7 K |
| | ConvBlock(64, 64) | 37 K |
| | ConvBlock(64, 64) | 37 K |
| | ConvBlock(64, 64) | 37 K |
| | Flatten | - |
| | ReLU | - |
| | Linear(576, 256) | 147 K |
| | ReLU | |
| | Linear(256, 128) | 32 K |

The overall number of parameters of the Prototypical Net we are using is around 292 K parameters. The loss of the Prototypical Net is applied on the output of the last fully connected layers (of size 128). For the computation of the samples diversity, we extract the features on the first fully-connected layer after the last convolutional layer (i.e., of size 256).

### Training details

The Prototypical Net is trained in a 1-shot 60-ways setting and tested on a 1-shot 20-ways setting. The size of the query set is always 1 for both training and testing phase. The model is trained during 80 epochs, with a batch size of 128. For training, we are using an Adam optimizer [28] with a learning rate of $1 \times 10^{-3}$ (all other parameters of the Adam optimizer are the default ones). We are scheduling the learning rate such that it is divided by 2 every 20 epochs.

At the end of the training, the training accuracy (evaluated on 1000 episodes) has reached 100% and the testing accuracy reaches a plateau at 96.55%.

## S2   More details on SimCLR

### S2.1   Architecture and Data Augmentation

The architecture we are using for SimCLR [10] is the exact same than the one used for Prototypical Net (see Table S1). In SimCLR, we also extract the features on the first fully-connected layer after the last convolutional layer (i.e., of size 256). The augmentations we use are randomly chosen among the 3 following transformations

- **Random resized crop:**  it crops random portion of the image and resizes it to a given size. 2 sets of parameters are used for this transformation: the scale and the ratio. The scale parameter specifies the lower and upper bounds for the random area of the crop. The ratio parameter specifies the lower and upper bounds for the random aspect ratio of the crop. Our scale range is (0.1, 0.9) and our ratio range is (0.8, 1.2).

- **Random affine transformation:**  it applies a random affine transformation of the image while keeping the center invariant. The affine transformation is a combination of a rotation

(from $-15°$ to $15°$), a translation (from $-5$ pixels to $5$ pixels), a zoom (with a ratio from $0.75$ to $1.25$) and a shearing (from $-10°$ to $10°$).

- **Random perspective transformation:** apply a scale distortion with a certain probability to simulate 3D transformations. The scale distortion we have chosen is $0.5$, and it is applied to the image with a probability of $50\%$

Please see the site `https://pytorch.org/vision/main/auto_examples/plot_transforms.html` for illustration of the transformations. Note that we have tried different settings for the augmentations (varying the parameters of the augmentations), and we have observed a very limited impact of those settings on the computation of the samples diversity (see S3.3 for more details).

## S2.2 Training details

Our SimCLR network is trained for $100$ epochs with a batch size of $128$. We used an RMSprop optimizer [25], with a learning rate of $10^{-3}$ (all other parameters of the RMSprop are the default ones).

## S3 Control experiments for the samples diversity computation

### S3.1 Comparing the supervised and the unsupervised settings for the computation of the samples diversity

To compare the unsupervised with the supervised setting, we have computed for all of the $150$ classes of the Omniglot testing set the samples diversity. We plot the samples diversity values for each category and for both settings in Fig. S1. We report a linear correlation coefficient $R^2 = 0.74$ and a Spearman rank order correlation $\rho = 0.85$ (see Table S2, first line). It does mean that the samples diversity, as computed with one of the setting, is strongly correlated both in terms of rank order and explained variance, with the samples diversity as computed with the other setting.

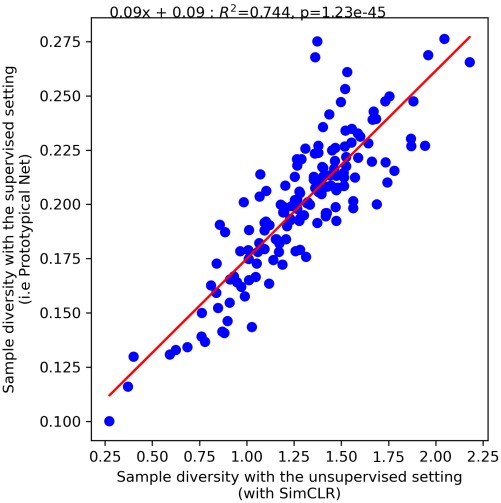

Figure S1: Comparison of the samples diversity computed by the supervised and the unsupervised settings. Each data point corresponds to a specific class in the Omniglot test set. Here, the samples diversity is computed applying the standard deviation (see Eq. 2) on SimCLR features (for x-axis) or on the features of Prototypical Net (for y-axis)

## S3.2 More control experiments on the effect of the dispersion measure

To make our analysis more robust we have conducted additional control experiments with different measures of dispersion. In Eq. 2 we have presented a classical measure of dispersion that is the standard deviation. Another measure of data dispersion is the pair-wise cosine distance among the

Table S2: Spearman rank order correlation for different settings

| Setting 1 | Setting 2 | Spearman correlation | p value |
|-----------|-----------|---------------------|---------|
| Proto. Net + Eq. 2 | SimCLR + Eq. 2 | 0.85 | $8.99 \times 10^{-43}$ |
| Proto. Net + Eq. 3 | SimCLR + Eq. 3 | 0.71 | $1.47 \times 10^{-24}$ |
| Proto. Net + Eq. 2 | Proto. Net + Eq. 3 | 0.73 | $1.19 \times 10^{-26}$ |
| SimCLR + Eq. 2 | SimCLR + Eq. 3 | 0.63 | $5.21 \times 10^{-18}$ |

samples belonging to the same class:

$$\sigma_{p_\theta}^j = \sum_{i=1}^{N} \sum_{\substack{k=1 \\ k>i}}^{N} \sqrt{2 - 2C(f(v_i^j), f(v_k^j))} \quad \text{s.t.} \quad v_i^j \sim p_\theta(\cdot | \tilde{x}^j) \quad \text{and} \quad C(u,v) = \frac{u \cdot v}{\|u\| \|v\|} \quad (3)$$

In Eq. 3, $C$ denotes the cosine similarity. In Fig. S2a, we plot the samples diversity for both feature extraction networks but with a dispersion measure based on the pairwise cosine distance as formulated in Eq. 3. We report a linear correlation of $R^2 = 0.57$ and a Spearman rank order correlation of $\rho = 0.71$ (see second line of Table S2). This control experiment suggests that even by using a different dispersion metric (i.e., the pairwise cosine distance), the 2 feature extraction networks produce samples diversity values that are heavily correlated. This strengthen our observation made in S3.1: the representations produced by the SimCLR and Prototypical Net are similar. Another interesting control experiment is to compare the impact of the dispersion measure on the samples diversity metric. To do so, we have compared the samples diversity computed with one feature extractor (either Prototypical Net in Fig. S2b or SimCLR in Fig. S2c) but for 2 different dispersion metrics (i.e., the standard deviation as formulated in Eq. 2 and the pairwise cosine distance as defined in Eq. 3). In both cases, we have a non negligible linear correlation (i.e., $R^2 > 0.44$) and a strong Spearman rank order correlation (i.e., $\rho > 0.63$, see third and fourth lines of Table S2). All these control experiments confirm that our computation of the samples diversity is robust to 1) the type of approach we used to extract the features and 2) the measure of dispersion we are using to compute the intraclass variability.

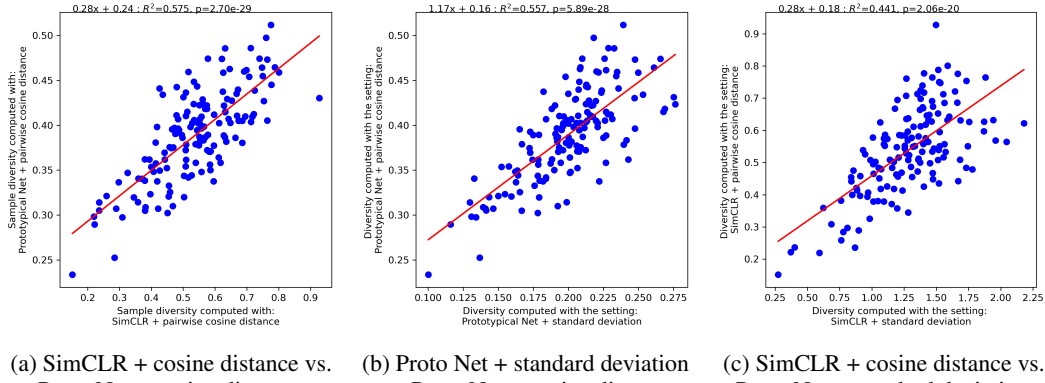

(a) SimCLR + cosine distance vs. Proto Net + cosine distance

(b) Proto Net + standard deviation vs. Proto Net + cosine distance

(c) SimCLR + cosine distance vs. Proto Net + standard deviation

Figure S2: Control experiments when the dispersion metric is the pairwise cosine distance (as defined in Eq. 3). Each point corresponds to a specific class of the Omniglot testing set. In a) we vary the feature extraction network, while keeping the same dispersion metric (i.e., the pairwise cosine distance). In b) and c), we fix the feature extraction network (Prototypical Net for b) and SimCLR for c)) and we vary the dispersion metric (standard deviation for the x-axis or the pairwise cosine distance for the y-axis).

### S3.3 Impact of the image augmentation on the diversity measure

To test the impact of the image augmentations on the SimCLR network we have trained 3 SimCLR networks with different augmentation levels.

- With moderate level of image augmentation. All the augmentations here are those described in section S2.

- With a low level of image augmentation. Here the scale of the random resized crop is varied from $0.05$ to $0.95$ and the crop ratio is ranging from $0.9$ to $1.1$. The rotation of the affine transformation is ranging from $-7\,\mathrm{deg}$ to $7\,\mathrm{deg}$, the translation from $-3$ pixels to $3$ pixels, the zoom from $0.9$ to $1.1$ and the shearing from $-5\,\mathrm{deg}$ to $5\,\mathrm{deg}$. The scale distortion applied to the image is $0.25$ (with a probability of $50\%$).

- With a high level of image augmentation. In this setting, the scale of the random resized crop is varied from $0.2$ to $0.8$ and the crop ratio is ranging from $0.6$ to $1.4$. The rotation of the affine transformation is ranging from $-30\,\mathrm{deg}$ to $30\,\mathrm{deg}$, the translation from $-10$ pixels to $10$ pixels, the zoom from $0.5$ to $1.5$ and the shearing from $-20\,\mathrm{deg}$ to $20\,\mathrm{deg}$. The scale distortion applied to the image is $0.75$ (with a probability of $50\%$).

In Fig. S3, we compare the samples diversity obtains for each category of the Omniglot testing set when we train the SimCLR network with moderate level of image augmentation and with a low level of image augmentation (see Fig. S3a), or with a high level of image augmentation (see Fig. S3b). We also report the Spearman correlation in Table S3.

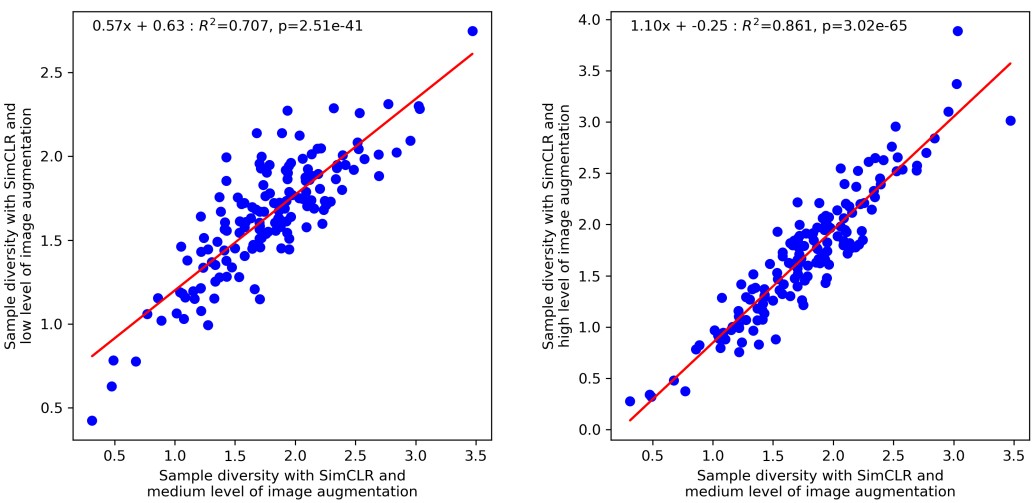

(a) SimCLR with moderate level of image augmentation versus SimCLR with low level of image augmentation

(b) SimCLR with moderate level of image augmentation versus SimCLR with high level of image augmentation

Figure S3: Control experiment to assess the impact of the level of image augmentation on the sample metric as evaluated as a standard deviation in the SimCLR feature space.

Table S3: Spearman rank order correlation for different settings

| Setting 1 | Setting 2 | Spearman correlation | p value |
|---|---|---|---|
| moderate augmentation | light augmentation | 0.79 | $1.7 \times 10^{-33}$ |
| moderate augmentation | strong augmentation | 0.90 | $1.45 \times 10^{-55}$ |

We observe a high linear correlation as well as a high Spearman rank order correlation between the tested settings. It suggests that the samples diversity is relatively independent to the level of image augmentations used during the SimCLR training.

### S3.4   T-SNE of the SimCLR and Prototypical Net latent space

In Fig. S4a and Fig. S4b, we show a t-SNE analysis of the feature space of Prototypical Net and SimCLR respectively. In Fig. S4a, the t-SNE analysis of the Prototypical Net feature space reveals

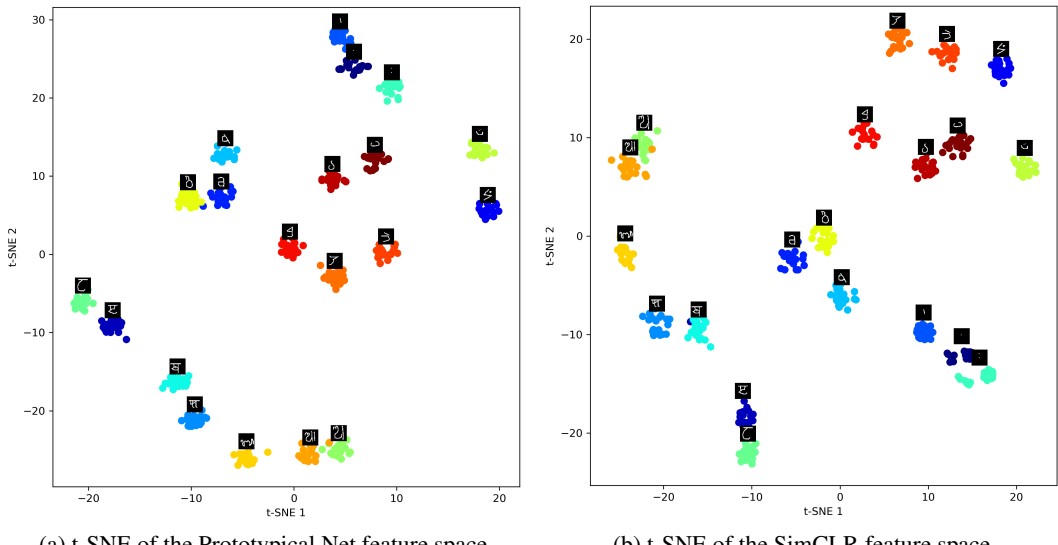

(a) t-SNE of the Prototypical Net feature space

(b) t-SNE of the SimCLR feature space

Figure S4: In these 2 figures the t-SNE analysis has been conducted on the 150 classes of the testing set of Omniglot. For the sake of clarity we show here a randomly selected subset of those classes (i.e., 20 classes).

a strong clustering of the samples belonging to the same class. Note that this phenomenon is not surprising as the loss of the Prototypical Net forces the samples belonging to the same class to be close in the feature space. More surprisingly, we also observe a clustering effect in the SimCLR t-SNE analysis (see Fig. S4b). Note that SimCLR is a fully unsupervised algorithm: there is no class information given to the algorithm. Consequently, the strong clustering effect we observe suggests that forcing the proximity between a sample and its augmented version is enough to retrieve the class information. This observation might explain why contrastive learning algorithms are in general so efficient in semi-supervised (or even unsupervised) classification tasks.

Figure S5: Concepts of the Omniglot test set, ranked by their diversity as computed with the unsupervised setting (i.e., SimCLR as a feature extractor and standard deviation for the dispersion measure). Here we linearly sub-sampled 30 of out of 150 concepts of the test set. Concepts are ranked in a increasing order (from low to high diversity). The samples in the red box are the prototypes, the rest of the line is composed with samples belonging to the same category.

Figure S6: Concepts of the Omniglot test set, ranked by their diversity as computed with the supervised setting (i.e., Prototypical Net as a feature extractor and standard deviation for the dispersion measure). Here we linearly sub-sampled 30 of out of 150 concepts of the test set. Concepts are ranked in a increasing order (from low to high diversity). The samples in the red box are the prototypes, the rest of the line is composed with samples belonging to the same category.

## S6 MAML architecture and training details

The architecture we have used for the MAML classifier is exactly the same used for the Prototypical Net (see Table S1). The only difference is the last fully-connected layer that is : Linear(256, 20). Indeed, as the MAML network is directly predicting the logits (and not a distance metric), the last layer needs to have the same dimension than the number of class of the experiment. In a 1-shot 20-way classification experiment, the number of classes is 20.

We have used a $2^{nd}$ order meta-learning scheme [18]. The outer-loop optimizer is an Adam optimizer with a learning rate of $10^{-3}$, and the inner-loop optimizer is a simple Stochastic Gradient Decent with a learning rate of $10^{-2}$. The number of inner loops is set to 5 during the training and to 10 during the testing. The number of tasks for each outer-loop is set to 4.

## S7 Control experiments: Comparing Prototypical Net and MAML

To rigorously compare MAML and Prototypical Net, we have conducted 2 types of control experiments. First we have verified whether the classification accuracy obtained for each class were ranked in the same order for both MAML and Prototypical Net. To do so, we have presented the same series of categorization tasks to both algorithms. The high Spearman rank coefficient ($\rho = 0.60$) indicates that both classifiers rank each category' classification accuracy similarly (see section S4).

To confirm this result, we have computed the correlation between the logits generated by both models. In the case of the MAML model, extracting the logits is straightforward. For Prototypical Net, we use the distance to prototypes as logits. This explain why both model's logits are anti-correlated: the MAML logits are the (un-normalized) probability of belonging to a given classes whereas the Prototypical Net logits correspond to the distance to the category (so the lower the distance, the higher the probability). We report a strong negative correlation ($r = -0.62$) between the logits of the MAML network and those of Prototypical Net (see section S4).

Table S4: Spearman rank order correlation for different settings

| Comparison | correlation type | correlation value | p value |
|---|---|---|---|
| MAML vs. Proto. Net (accuracy) | Spearman | 0.60 | $4.24 \times 10^{-15}$ |
| MAML vs. Proto. Net (logits) | Pearson | -0.62 | $2.63 \times 10^{-19}$ |

## S8 Architecture and training details of the VAE-STN

### S8.1 Architecture of the VAE-STN

The VAE-STN is a sequential VAE that allows for the iterative construction of a complex image [42]. A pseudo-code of the algorithm is described in Algo 1. At each iteration, the algorithm focuses its attention on a specific part of the image ($x$), the prototype ($\tilde{x}$) and the residual image ($\hat{x}$) using the Reading Spatial Transformer Network ($STN_r$). Then the extracted patch is passed to an encoding network (EncBlock) to transform it into a latent variable. This latent variable is concatenated to a patch extracted from the prototype and then passed to the RecBlock network. The produced hidden state is first passed to DecBlock to recover the original patch, and then to the $STN_w$ to replace and rescale the patch into the original image. The LocNet network is used to learn the parameter of the affine transformation we used in the STN. Note that the affine parameters used in $STN_w$ are simply the inverse of those used in $STN_r$.

---

**Algorithm 1** Pseudo-code of the VAE-STN

---

**Input:** image: **x**, prototype: $\tilde{x}$

$c \leftarrow \mathbf{0}$
$\boldsymbol{\theta_1} \leftarrow [[1,0,0],[0,1,0]]$
$h_1 \leftarrow \mathbf{0}$
**for** $i = 1$ to $N_{steps}$ **do**
$\quad \hat{x} = x - \text{sigmoid}(c)$
$\quad r,\ \hat{r},\ \tilde{r} = STN_r(\boldsymbol{\theta_t}, x),\ STN_r(\boldsymbol{\theta_t}, \hat{x}),\ STN_r(\boldsymbol{\theta_t}, \tilde{x})$
$\quad r \leftarrow [r, \hat{r}, \tilde{r}, h_t]$
$\quad \mu, \sigma = EncBlock(r)$
$\quad z = \mu + \epsilon \sigma$ with $\epsilon \sim \mathcal{N}(0,1)$
$\quad z \leftarrow [z, \tilde{r}]$
$\quad p = DecBlock(h_t)$
$\quad c \leftarrow c + STN_w(\boldsymbol{\theta_t^{-1}}, p)$
$\quad h_{t+1} \leftarrow RecBlock(z, h_t)$
$\quad \boldsymbol{\theta_t} + 1 \leftarrow LocNet(h_{t+1})$
**end for**

---

The STN modules take 2 variables in input: an image (or a patch in the case to the $STN_w$) and a matrix (3×2) describing the parameters of the affine transformation to apply to the input image [26]. All other modules are made with MLPs networks, and are described in Table S5. In the Table S5 we use the following notations:

- $s_z$: This the size of the latent space. In the base architecture, we set $s_z = 80$.

- $s_{LSTM}$: This is the size of the output of the Long-Short Term Memory (LSTM) unit. In the base architecture, we set $s_{LSTM} = 400$

- $s_r$: This is the resolution of the patches extracted by the Spatial Transformer Net (STN) during the reading operation. In the base architecture we set $s_r = 15$.

- $s_{loc}$: This is the number of neurons used at the input of the localization network. In the base architecture, we set $s_{loc} = 100$

- $s_w$: This is the resolution of the patch passed to the the STN network for the writing operation. In the base architecture $s_w = 15$.

For the base architecture we used $N_{steps} = 60$. The base architecture of the VAE-STN has 6.2 millions parameters. For more details on the loss function, please refer to [42].

### S8.2 Training details of the VAE-STN

The VAE-STN is trained for 500 epochs, with batches of size 128. We use an Adam optimizer with a learning rate of $1 \times 10^{-3}$ and $\beta_1 = 0.9$. All other parameters are the default Pytorch parameters. To avoid training instabilities we clip the norm of the gradient to 5. The learning rate was divided by 2 when the evaluation loss has not decreased for 10 epochs (reduce on plateau strategy).

Table S5: Description of the VAE-STN architecture

| Network | Layer | # params |
|---|---|---|
| EncBlock($s_r$, $s_{LSTM}$, $s_z$) | Linear($3 \times s_r^2 + s_{LSTM}$ , 1024) | $(3 \times s_r^2 + s_{LSTM}) \times 1024)$ + 1024 |
| | ReLU | |
| | Linear(1024, 1024) | 1050 K |
| | ReLU | |
| | Linear(1024, 512) | 524 K |
| | ReLU | - |
| | Linear(512, 128) | 65 K |
| | ReLU | - |
| | Linear(128, $2 \times s_z$) | $256 \times s_z + 2 \times s_z$ |
| LocNet($s_{loc}$) | Linear($s_{loc}$, 64) | $s_{loc} \times 64 + 64$ |
| | ReLU | - |
| | Linear(64, 32) | 2 K |
| | ReLU | - |
| | Linear(32, 6) | 0.2 K |
| DecBlock($s_{LSTM}$, $s_{loc}$, $s_w$) | Linear($s_{LSTM}$ - $s_{loc}$, 1024) | $(s_{LSTM}$ - $s_{loc}) \times 1024 + 1024$ |
| | ReLU | - |
| | Linear(1024, 512) | 525 K |
| | ReLU | - |
| | Linear(512, 256) | 131 K |
| | ReLU | - |
| | Linear(256, $s_w^2$) | $256 \times s_w^2 + s_w^2$ |
| RecBlock($s_z$, $s_r$, $s_{LSTM}$) | LSTMCell($s_z + s_r^2$, $s_{LSTM}$) | $4 \times \left(s_z + s_r^2\right) \times s_{LSTM}$ $+ s_{LSTM}^2 + s_{LSTM}\right)$ |
| VAE-STN | EncBlock(15 , 800, 80) | $3,172$ K |
| | RecBlock(80, 15, 800) | $1,600$ K |
| | DecBlock(400, 100, 15) | $1,431$ K |
| | LocNet(100) | $8.7K$ |

## S8.3   VAE-STN samples

Figure S7: Sampled generated by the **VAE-STN**. All the prototypes used to condition the generative model are in the red frame. The 30 concepts has been randomly sampled (out of 150 concepts) from the Omniglot test set. The lines are composed with 20 samples that has been generated by the VAE-STN.

## S9 Architecture and training details of the Neural Statistician

**Architecture**

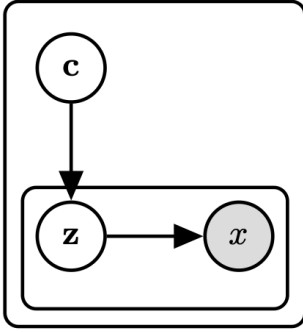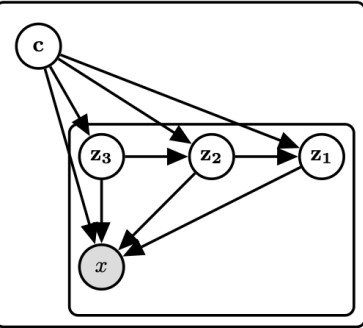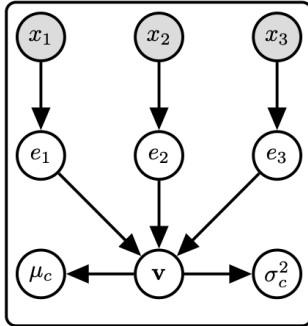

Figure S8: *Left*: basic hierarchical model, where the plate encodes the fact that the context variable $c$ is shared across each item in a given dataset. *Center*: full neural statistician model with three latent layers $z_1, z_2, z_3$. Each collection of incoming edges to a node is implemented as a neural network, the input of which is the concatenation of the edges' sources, the output of which is a parameterization of a distribution over the random variable represented by that node. *Right*: The statistic network, which combines the data via an exchangeable statistic layer. The above figures were obtained from [15]

Table S6 describes the base architecture of the Neural Statistician model adopted from [19] which is a close approximation of [15]. We make minor changes in the network architecture to accommodate the higher input image size of $50 \times 50$ of the Omniglot dataset. The Neural Statistician model is composed of the following sub-networks:

- Shared encoder $x \mapsto h$: An instance encoder $E$ that takes each individual datapoint $x_i$ to a feature representation $h_i = E(x_i)$.
- Statistic network $q(c|D, \phi) : h_1, ..., h_k \mapsto \mu_c, \sigma^2{}_c$: A pooling layer that aggregates the matrix $(h_1, ..., h_k)$ to a single pre-statistic vector $v$. [15] uses sample mean for their experiments. Which is followed by a post-pooling network that takes $v$ to a parametrization of a Gaussian.
- Inference network $q(z|x, c, \phi) : h, c \mapsto \mu_z, \sigma^2{}_z$: Inference network gives an approximate posterior over latent variables.
- Latent decoder network $p(z|c; \theta) : c \mapsto \mu_z, \sigma^2{}_z$
- Observation decoder network $p(x|c, z; \theta) : c, z \mapsto \mu_x$

The overall number of parameters of the base model (which has the same architecture as used in [15]) for the Neural Statistician we are using is around 7.48M parameters.

**Training details**

The Neural Statistician is trained for 300 epochs, with batch size of 32 and learning rate of $1 \times 10^{-3}$. We adopt the same setting of the Neural Statistician as used in [15] for the omniglot dataset. We constructed context sets by splitting each class into datasets of size 5 while training, and use a single out-of-distribution exemplar while testing. As discussed in the paper, we create new classes by reflecting and rotating characters. We based our implementation from https://github.com/georgosgeorgos/hierarchical-few-shot-generative-models and https://github.com/comRamona/Neural-Statistician.

**Intuition about context integration in the Neural-Statistician**

In the Neural Statistician, the context correspond to the samples used during training, to evaluate the statistics of a specific category (i.e. a concept). In practice, we pass to the network different samples representing the same concept and we vary the number of these samples (from 2 to 20 in the experiment described in section 4.2). Intuitively, with more context samples for a given category,

Table S6: Description of the Neural Statistician Architecture

| Network | Layer | # params |
|---|---|---|
| ConvBlock(In$_c$, Out$_c$, stride) | Conv2d(In$_c$, Out$_c$, stride, 3, padding=1) | In$_c$ × Out$_c$ × 3 × 3 + Out$_c$ |
| | BatchNorm2d(Out$_c$), ELU | 2 x Out$_c$ |
| FcBlock(In, Out) | Linear(In, Out) | In × Out |
| | BatchNorm1d(Out), ELU | - |
| DeConvBlock(In$_c$, Out$_c$) | ConvTranspose2d(In$_c$, Out$_c$, 2, 2) | In$_c$ × Out$_c$ × 3 × 3 + Out$_c$ |
| | BatchNorm2d(Out$_c$), ELU | 2 x Out$_c$ |
| Shared encoder | ConvBlock(1, 32, 1)
ConvBlock(32, 32, 1)
ConvBlock(32, 32, 2)
ConvBlock(32, 64, 1)
ConvBlock(64, 64, 1)
ConvBlock(64, 64, 2)
ConvBlock(64, 128, 1)
ConvBlock(128, 128, 1)
ConvBlock(128, 128, 2)
ConvBlock(128, 256, 1)
ConvBlock(256, 256, 1)
ConvBlock(256, 256, 2) | 1,958,400 |
| Statistic network | FcBlock(256*4*4, 256)
average pooling within each dataset
2× FcBlock(256, 256)
Linear(256, 512), BatchNorm1d(1) to $\mu_c$, $\log \sigma^2{}_c$ | 1,445,122 |
| Inference network | FcBlock(256, 256) $\mapsto h$
FcBlock(512, 256) $\mapsto c$
combine c and h, ELU
Residual Block{3× FcBlock(256, 256)}
Linear(256, 32), BatchNorm1d(1) to $\mu_z$, $\log \sigma^2{}_z$ | 408,610 |
| Latent decoder network | Linear(512, 256) $\mapsto c$, ELU
Residual Block{3× FcBlock(256, 256)}
Linear(256, 32), BatchNorm1d(1) to $\mu_z$, $\log \sigma^2{}_z$ | 342,818 |
| Observation decoder network | FcBlock(512, 256) $\mapsto z$
FcBlock(512, 256) $\mapsto c$
combine z and c, ELU
FcBlock(256, 256*4*4)
ConvBlock(256, 256, 1)
ConvBlock(256, 256, 1)
DeConvBlock(256, 256)
ConvBlock(256, 128, 1)
ConvBlock(128, 128, 1)
DeConvBlock(128, 128)
ConvBlock(128, 64, 1)
Conv2d(64, 64, 4, 1, 0)
DeConvBlock(64, 64)
ConvBlock(64, 32, 1)
Conv2d(32, 32, 2, 1, 0)
DeConvBlock(32, 32)
Conv2d(32, 1, 1) | 3,324,673 |

it becomes easier for the network to identify the properties and features that are crucial to define a given handwritten letter (which results in a higher recognizability but leaves less room for diversity).

## S9.1 Neural statistician samples

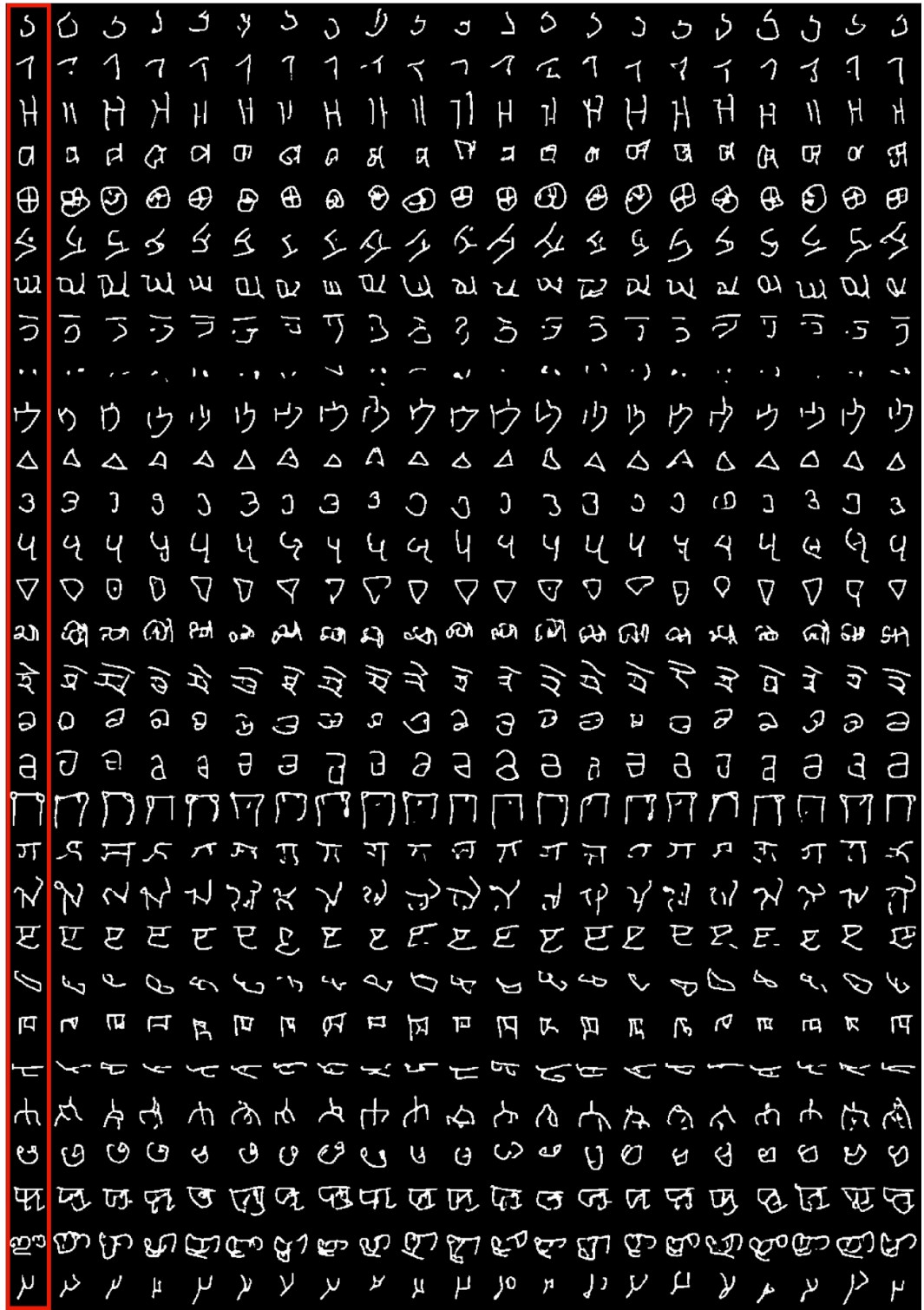

Figure S9: Sampled generated by the neural statistician network (**VAE-NS**). All the prototypes used to condition the generative model are in the red frame. The 30 concepts has been randomly sampled (out of 150 concepts) from the Omniglot test set. The lines are composed with 20 samples that has been generated by the VAE-NS.

## S10    Architecture and training details of the DA-GAN based on U-Net (DA-GAN-UN)

**Architecture**

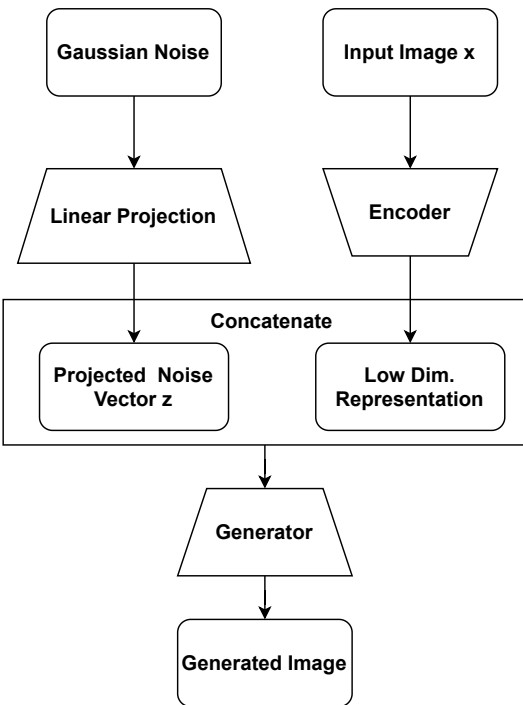

Figure S10: DAGAN Generator: The generator is composed of an encoder projecting the input image to a lower dimensional manifold. A random gaussian noise vector is transformed and concatenated with the bottleneck vector. The resulting vector is passed through the decoder (generator), which outputs the augmented image.

Table S7 describes the base architecture of the DA-GAN-UN's Generator model adopted from [1]. We have modified the architecture of the DA-GAN-UN model such that it can accommodate a higher input image size $50 \times 50$. Also, we reduced the number of trainable parameters in the original DA-GAN-UN architecture to have a fair comparison with other few-shot models. Following are the notations used in Table S7:

- $s_z$: This is the size of the latent space. In the base architecture, we set $s_z = 128$
- Generator $G(x, z)$: A generator network that takes data points and Gaussian noise as input, and generate new samples.

The base architecture of the DAGAN model we are using in our experiments has around 6.8 million parameters.

**Training details**

The DA-GAN-UN model was trained for 30 epochs, with batches of size 32. We use an Adam optimizer with a learning rate of $1 \times 10^{-4}$ and $\beta_1 = 0.9$. We update our generator after every 5 updates of discriminator. We based our implementation from https://github.com/amurthy1/dagan_torch

Table S7: Description of the Data Augmentation GAN Architecture

| Network | Layer | # params |
|---|---|---|
| ConvBlock(In$_c$, Out$_c$, s$_l$) | Conv2d(In$_c$, Out$_c$, 3, stride=s$_l$, padding=1) 
 LeakyReLU(0.2), BatchNorm2d(Out$_c$) | Out$_c$ × (In$_c$ × 3 × 3 + 1) 
 2 x Out$_c$ |
| DeConvBlock(In$_c$, Out$_c$, s$_l$) | ConvTranspose2d(In$_c$, Out$_c$, 3, stride=s$_l$, padding=1) 
 LeakyReLU(0.2), BatchNorm2d(Out$_c$) | Out$_c$ × (In$_c$ × 3 × 3 + 1) 
 2 x Out$_c$ |
| EncoderBlock(In$_p$, In$_c$, Out$_c$) | ConvBlock(In$_p$, In$_p$) 
 ConvBlock(In$_c$ + In$_p$, Out$_c$) 
 Conv2d(In$_c$+ Out$_c$, Out$_c$) 
 Conv2d(In$_c$+ 2 × Out$_c$, Out$_c$) | |
| DecoderBlock(In$_p$, In$_c$, Out$_c$) | DeConvBlock(In$_p$, In$_p$, 1) 
 ConvBlock(In$_c$+In$_p$, Out$_c$, 1) 
 DeConvBlock(In$_p$, In$_p$, 1) 
 ConvBlock(In$_c$ + In$_p$ + Out$_c$, Out$_c$, 1) 
 DeConvBlock(In$_c$ + 2 × Out$_c$, Out$_c$, 1) | |
| Generator(s$_z$) | ConvBlock(1, 64, 2) 
 EncoderBlock(1, 64, 64) 
 EncoderBlock(64, 64, 128) 
 EncoderBlock(128, 128, 128) 
 Linear(s$_z$, 4×4×8) 
 DecoderBlock(0, 136, 64) 
 Linear(s$_z$, 7×7×4) 
 DecoderBlock(128, 260, 64) 
 Linear(s$_z$, 13×13×2) 
 DecoderBlock(128, 194, 64) 
 DecoderBlock(64, 128, 64) 
 DecoderBlock(64, 65, 64) 
 ConvBlock(64, 64, 1) 
 ConvBlock(64, 64, 1) 
 Conv2d(64, 1, 3, stride=1, padding=1) | 6,813,857 |

## S10.1  DA-GAN-UN samples

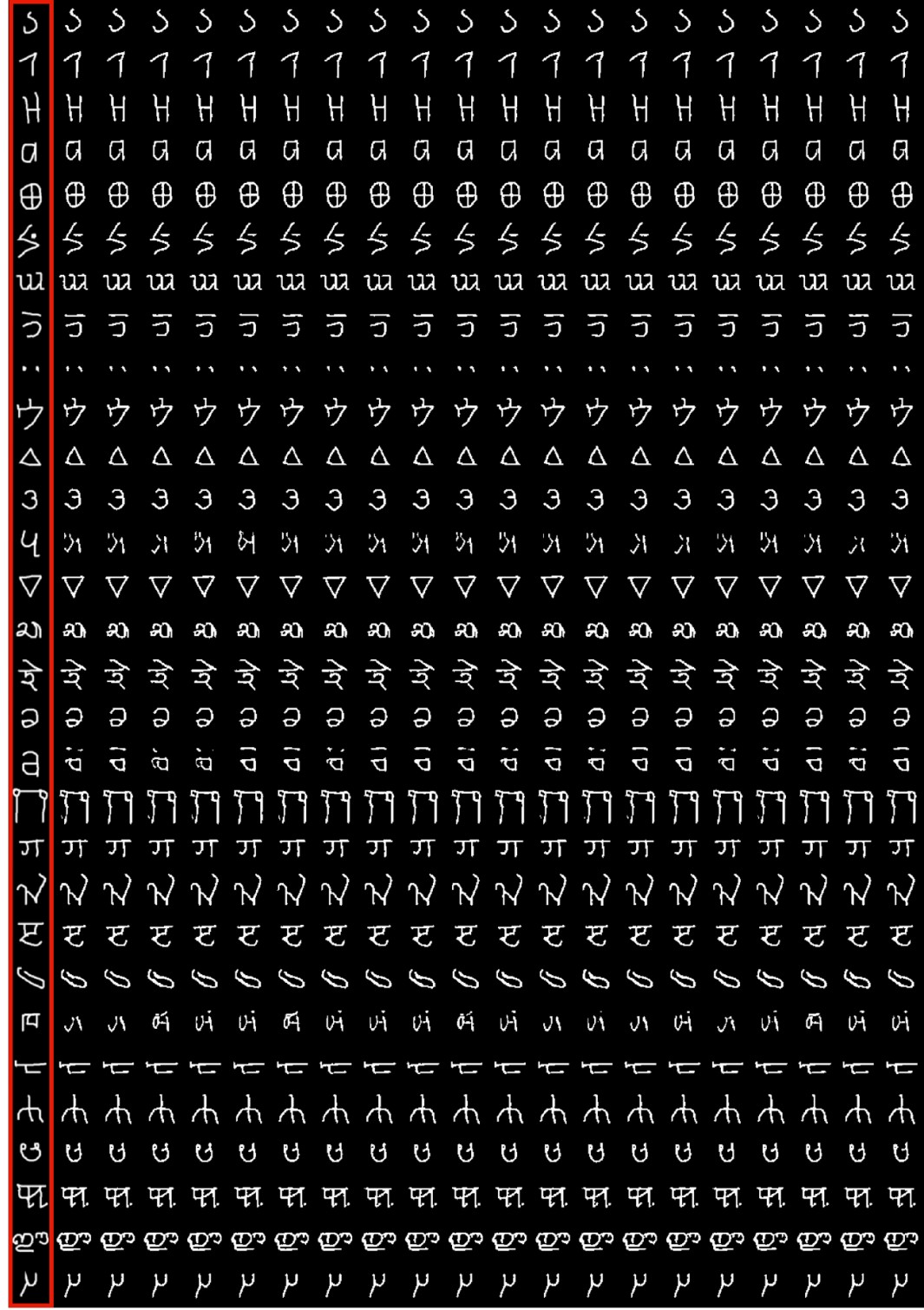

Figure S11: Sampled generated by the Data Augmentation GAN with U-Net architecture (**DA-GAN-UN**). All the prototypes used to condition the generative model are in the red frame. The 30 concepts has been randomly sampled (out of 150 concepts) from the Omniglot test set. The lines are composed with 20 samples that has been generated by the DA-GAN-UN.

## S11 Architecture and training details of the DA-GAN based on ResNet (DA-GAN-RN)

**Architecture**

We use the same base architecture of **DA-GAN-UN**, except we remove the skip connections between the contracting path (encoder) and the expansive path (decoder). [1] used a combination of UNet and ResNet in their results, in **DA-GAN-RN** we consider only a ResNet type architecture.

**Training details**

Refer S10 for training details.

## S11.1  DA-GAN-RN samples

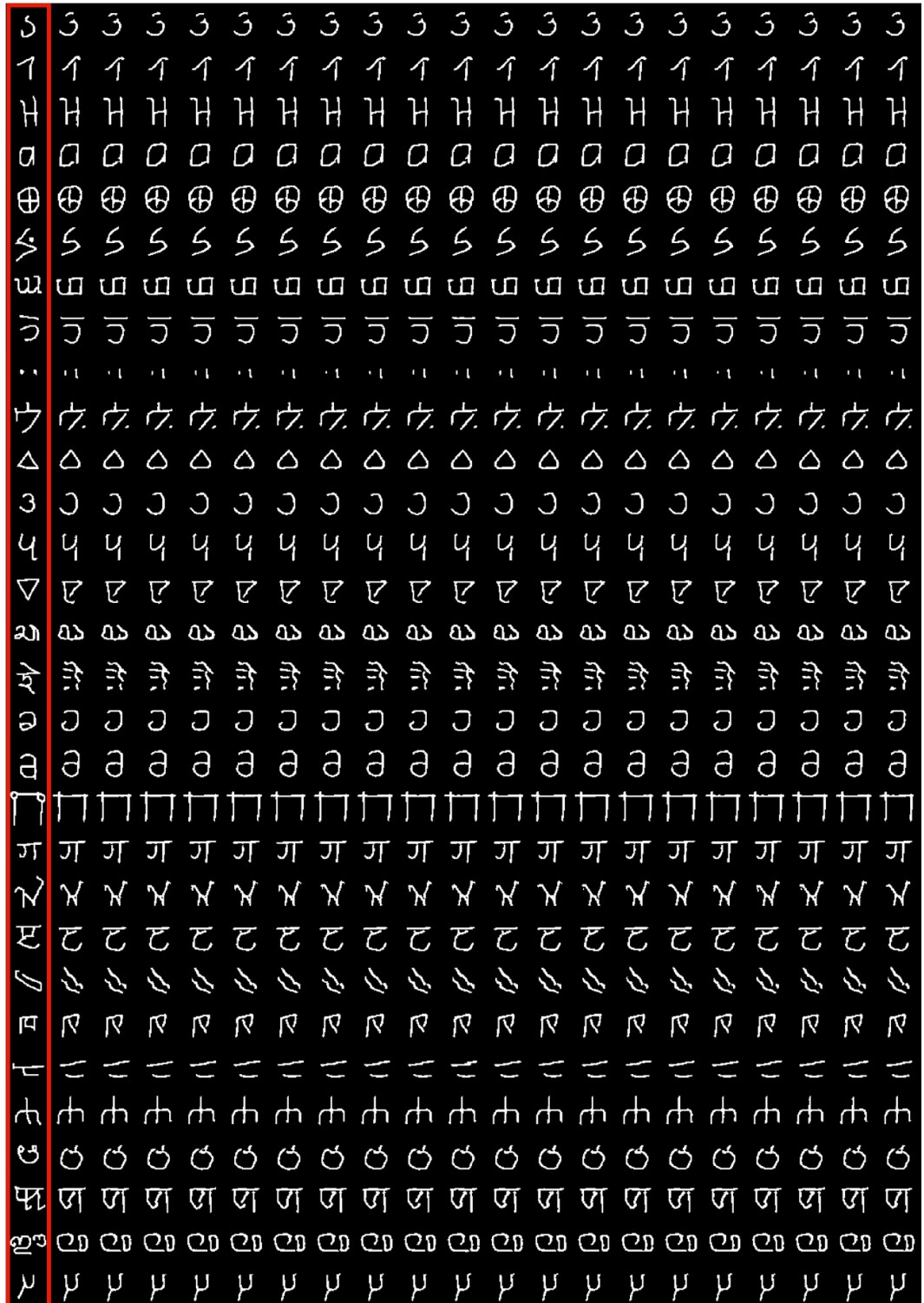

Figure S12: Sampled generated by the Data Augmentation GAN with ResNet architecture (**DA-GAN-RN**). All the prototypes used to condition the generative model are in the red frame. The 30 concepts has been randomly sampled (out of 150 concepts) from the Omniglot test set. The lines are composed with 20 samples that has been generated by the DA-GAN-RN.

## S12    Effect of the number of context samples on the diversity/recognizability framework

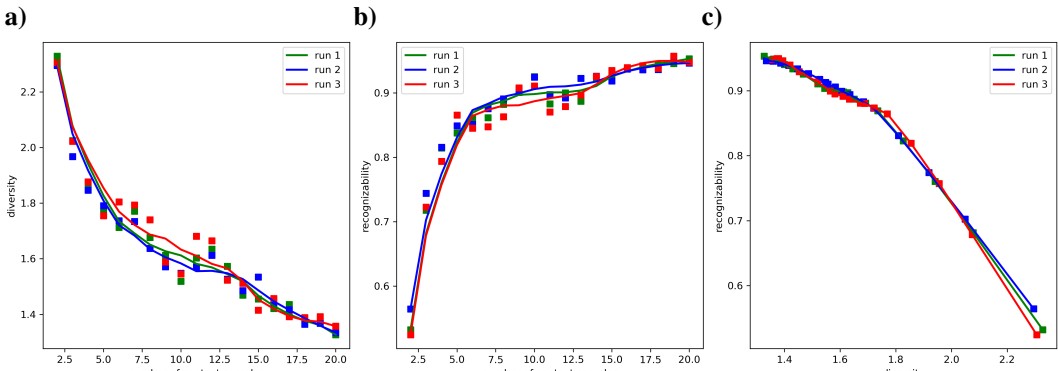

Figure S13: Effect of the number of context samples on the diversity/recognizability framework for 3 different runs. (**a**) Effect of the number of context samples on the diversity. (**b**) Effect of the number of context samples on the recognizability. (**c**) Simultaneous evolution of diversity and recognizability when ones varies the number of context samples from 2 to 20.

We observe a monotonic decrease of the diversity and a monotonic increase of the recognizability when the number of context samples increases. We vary the number of context samples from 2 to 20. This experiment has been conducted with 3 different seeds (i.e., different network initialization), represented with red, green and blue data points, respectively. For each seed, we report 19 data points. To highlight the trend in the diversity-recognizability space, we have smoothed the curves in Fig. S13a and Fig. S13b, using a Savitzky-Golay filter (second order, window size of 7).

## S13    Effect of the number of attentional steps on the diversity/recognizability framework

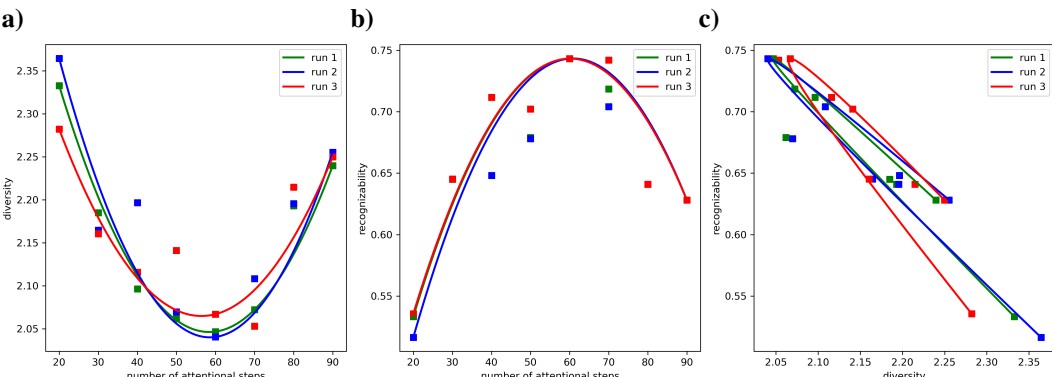

Figure S14: Effect of the number of attentional steps on the diversity/recognizability framework for 3 different runs. (**a**) Effect of the number of attentional steps on the diversity. (**b**) Effect of the number of attentional steps on the recognizability. (**c**) Simultaneous evolution of diversity and recognizability when one varies the number of attentional steps from 20 to 90

In this experiment, we have varied the number of attentional steps from 20 to 90. Note that we could not go below 20 attentional steps to make sure the attentional process is fully covering the entire image. We did not go over 90 attentional steps because we faced some training instabilities beyond this point. We observe a non-monotonic evolution of the diversity and the recognizability with the increase of the number of attentional steps. This experiment has been conducted with 3 different seeds (i.e., different network initialization), represented with red, green and blue data points, respectively. For each seed we report 8 data points. In order to properly assess the type of parametric curves that govern the evolution of the diversity-recognizability space when one varies the number of attentional steps, we have used a least curve fitting method [21]. This method involves finding the

best polynomial fit (second order in our case) for the 3 curves (Fig. S14a, b and c) simultaneously. This method is iteratively refining all the fits to minimize the sum of all least square error.

## S14  Mathematical formulation of the ELBO

Let us consider a dataset $\boldsymbol{X} = \{\boldsymbol{x}^{(i)}\}_{i=1}^{N}$ composed of $N$ i.i.d samples of a random variable $\boldsymbol{x}$. We assume that $\boldsymbol{x}$ is generated by some random process involving an unobserved random variable $\boldsymbol{z}$. The latent variable $\boldsymbol{z}$ is sampled from a Gaussian distribution (see Eq. 5). The mean of the likelihood is parametrized by $\boldsymbol{\mu}_\theta$ (in which $\theta$ denotes the parameters) and its variance is considered constant.

$$\boldsymbol{x} \sim p_\theta(\boldsymbol{x} \mid \boldsymbol{z}) \quad \text{s.t} \quad p_\theta(\boldsymbol{x} \mid \boldsymbol{z}) = \mathcal{N}\big(\boldsymbol{x}; \boldsymbol{\mu}_\theta(\boldsymbol{z}), \boldsymbol{\sigma}_x^2\big) \tag{4}$$

$$\boldsymbol{z} \sim p(\boldsymbol{z}) \quad \text{s.t} \quad p(\boldsymbol{z}) = \mathcal{N}\big(\boldsymbol{z}; \boldsymbol{\mu}_p, \boldsymbol{\sigma}_p^2\big) \tag{5}$$

The Variational Auto Encoder is optimized by maximizing the Evidence Lower Bound (ELBO), as formalized in its simplest form in Eq. 6:

$$ELBO(\boldsymbol{x}, \theta, \phi) = \mathbb{E}_{q_\phi(\boldsymbol{z}|\boldsymbol{x})}[\log p_\theta(\boldsymbol{x} \mid \boldsymbol{z})] - \beta \, \text{KL}\big(q_\phi(\boldsymbol{z} \mid \boldsymbol{x}) \| p(\boldsymbol{z})\big) \tag{6}$$

One could observe that the $\beta$ coefficient is tuning the importance of the prior (through the KL). If $\beta > 1$, then the latent space will be forced to be closer to the prior distribution but will attenuate the weight of the reconstruction loss. Such a scenario tends to improve the disentanglement of the latent space [24]. On the contrary, if $\beta$ is low, then the reconstruction loss (i.e., $\mathbb{E}_{q_\phi(\boldsymbol{z}|\boldsymbol{x})}[\log p_\theta(\boldsymbol{x} \mid \boldsymbol{z})]$) will take over, and then the latent space will be less regularized. Note that in the extreme case where $\beta = 0$, the VAE becomes an auto-encoder.

The ELBO loss can be updated to include a latent variable encoding for the context $\boldsymbol{c}$ as in the **VAE-NS**. In this formulation, the context corresponds to a dataset $D$ (see Eq. 7):

$$ELBO(\boldsymbol{x}, \theta, \phi) = \mathbb{E}_{q_\phi(\boldsymbol{c}|D)}\Big[ \sum_{\boldsymbol{x} \in D} \mathbb{E}_{q_\phi(\boldsymbol{z}|\boldsymbol{c},\boldsymbol{x})}[\log p_\theta(\boldsymbol{x} \mid \boldsymbol{z})] - \beta \, \text{KL}\big(q_\phi(\boldsymbol{z} \mid \boldsymbol{c}, \boldsymbol{x}) \| p(\boldsymbol{z} \mid \boldsymbol{c})\big) \Big] \tag{7}$$
$$- \text{KL}\big(q_\phi(\boldsymbol{z} \mid D) \| p(\boldsymbol{c})\big)$$

The ELBO could also be extended to include a sequential generative process as in the **VAE-STN**. In this case, the latent variable $\boldsymbol{z}$ is time-indexed and is now a sequence of random variables denoted $(\boldsymbol{z_1}, .., \boldsymbol{z_T})$. In Eq. 8, $\boldsymbol{z_{<k}}$ indicates the collection of all latent variables from step $t = 1$ to $t = k$.

$$ELBO(\boldsymbol{x}, \theta, \phi) = \mathbb{E}_{q_\phi(\boldsymbol{z_1},..,\boldsymbol{z_T}|\boldsymbol{x})}[\log p_\theta(\boldsymbol{x} \mid \boldsymbol{z_1}, .., \boldsymbol{z_T})] - \beta \, \text{KL} \sum_{k=1}^{T} \big(q_\phi(\boldsymbol{z_k} \mid \boldsymbol{z_{<k}}, \boldsymbol{x}) \| p(\boldsymbol{z_k})\big) \tag{8}$$

## S15  Effect of the beta coefficient on the diversity/recognizability framework

In this experiment, we have varied the value of the $\beta$ coefficient from 0.25 to 4 for the **VAE-STN** and from 0.25 to 5 for **VAE-NS** model. This experiment has been conducted with 3 different seeds (i.e., different network initialization), represented with red, green and blue data points, respectively. For the **VAE-STN** and for each seed, we have collected 16 data points (see Fig. S15), and 20 for the **VAE-NS** (see Fig. S16). We use a similar method than in S13 to find a polynomial fit (second order in our case) of the curves shown in Fig. S15a, b, and c and Fig. S16a, b, and c. We report a quasi-monotonic decline of the diversity when the beta value is increased (see Fig. S15a and Fig. S16a). In contrast, the recognizability follows a parabolic relationship when the beta value is increased. For the **VAE-STN**, the maximum recognizability ($\approx 80\%$) is reached for a $\beta$ value of 2.25 (see Fig. S15b). For the **VAE-NS**, the maximum recognizability ($\approx 91\%$) is reached for a $\beta$ value of 3 (see Fig. S16b). Even if the change of amplitude in recognizability and in diversity is larger for the **VAE-STN** than for **VAE-NS**, the shapes of the curves are very similar.

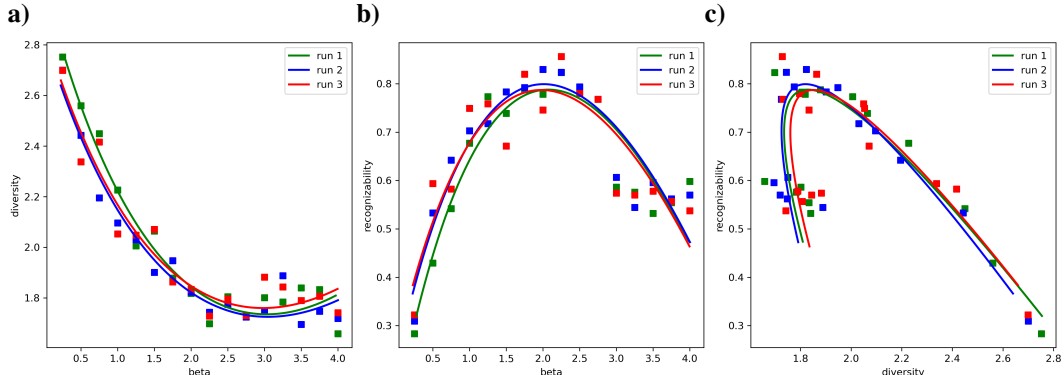

Figure S15: Effect of varying $\beta$ in the **VAE-STN** on the diversity/recognizability framework for 3 different runs. (**a**) Effect of $\beta$ on the diversity. (**b**) Effect of $\beta$ on the recognizability. (**c**) Parametric curve recognizability versus diversity when one varies $\beta$ from 0.25 from to 4.

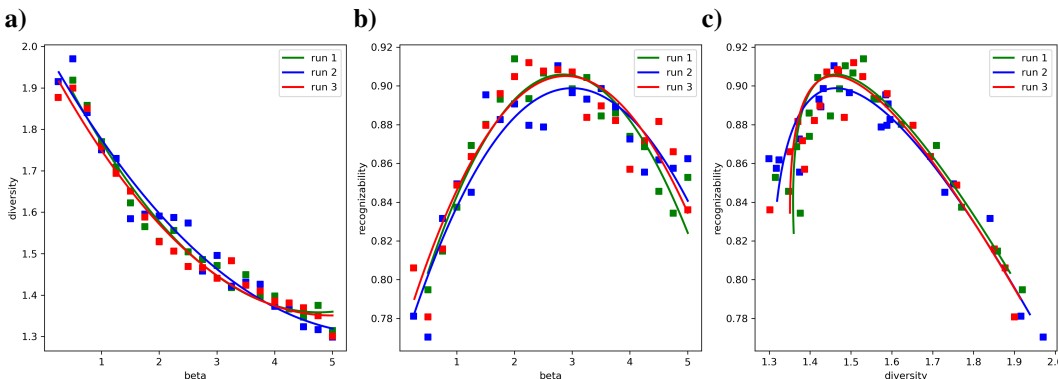

Figure S16: Effect of varying $\beta$ in the **VAE-NS** on the diversity/recognizability framework for 3 different runs. (**a**) Effect of $\beta$ on the diversity. (**b**) Effect of $\beta$ on the recognizability. (**c**) Parametric curve recognizability versus diversity when one varies $\beta$ from 0.25 from to 5.

## S16 Effect of the size of the latent space on the diversity/recognizability framework

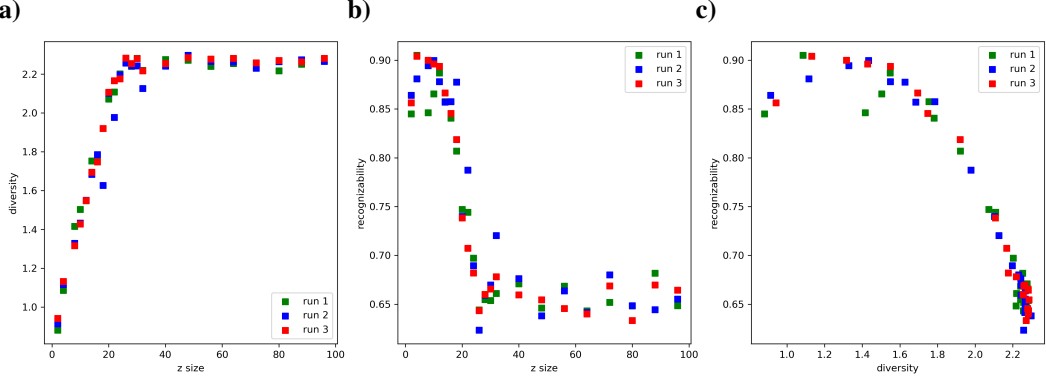

Figure S17: Effect of varying the size of the latent vector ($z$) in the **VAE-NS** on the diversity/recognizability framework for 3 different runs. (**a**) Effect of latent size on the diversity. (**b**) Effect of the latent size on the recognizability. (**c**) Parametric curve recognizability versus diversity when one varies $\beta$ from 5 from to 100.

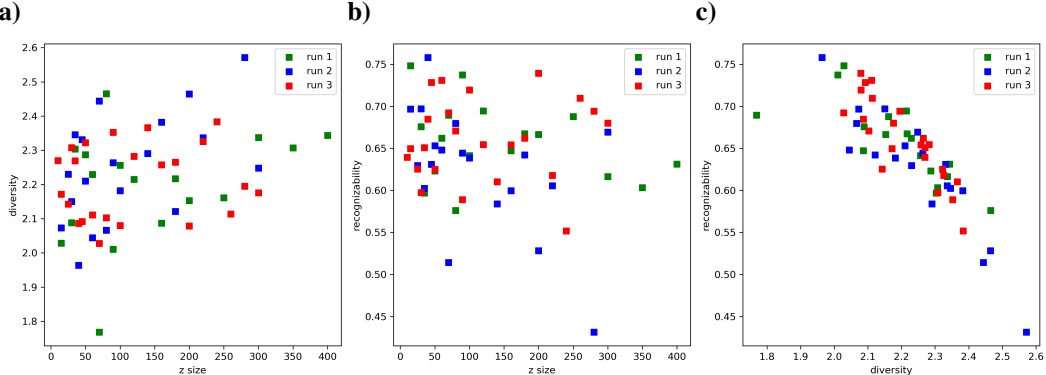

Figure S18: Effect of varying the size of the latent vector ($z$) in the **VAE-STN** on the diversity/recognizability framework for 3 different runs. (**a**) Effect of latent size on the diversity. (**b**) Effect of the latent size on the recognizability. (**c**) Parametric curve recognizability versus diversity when one varies $\beta$ from 5 from to 400.

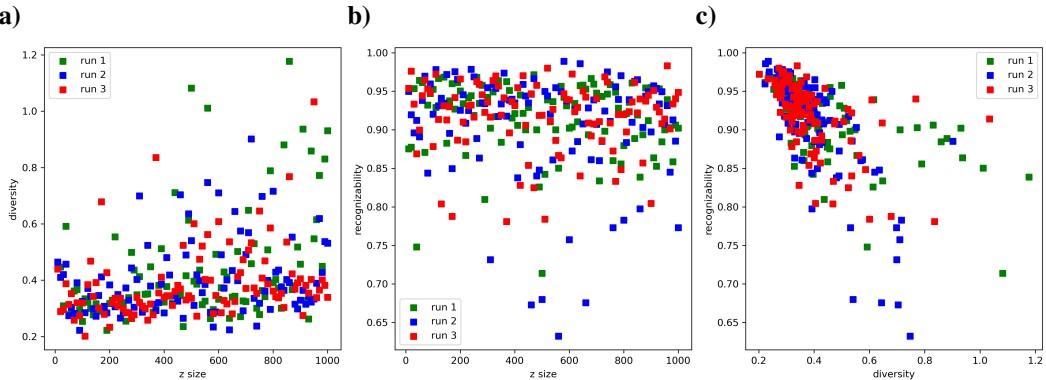

Figure S19: Effect of varying the size of the latent vector ($z$) in the **DA-GAN-UN** on the diversity/recognizability framework for 3 different runs. (**a**) Effect of latent size on the diversity. (**b**) Effect of the latent size on the recognizability. (**c**) Parametric curve recognizability versus diversity when one varies $\beta$ from 10 from to 1000.

## S17 Overfitting of standard classifier in low-data regime

Omniglot is a dataset composed of images representing 1,623 classes of handwritten letters and symbols (extracted from 50 different alphabets) with just 20 samples per class. This low number of samples per class makes Omniglot very different from other datasets (e.g. MNIST, CIFAR10...). In such a low-data regime, standard deep learning classifiers are known to overfit to the training data [7] resulting in poor generalization performance. In this section we provide experimental confirmation of such a phenomenon.

We have trained 3 different classifiers, all having a similar architecture (the architecture is described in Table S1):

- **A standard classifier**. For this classifier, the last linear layer has been changed to have an output activation of size 1623. Said differently, the layer entitled "Linear(256, 128)" in Table S1 has been replaced by "Linear(256, 1623)". We have trained this classifier using 18 samples per class of the Omniglot dataset. The testing set is composed of the 2 remaining samples per class. To summarize, the training set is composed of $29,214$ samples ($1623 \times 18$) and the training set is composed of 3246 samples ($1623 \times 2$). This classifier is trained using a standard back-propagation on a cross-entropy loss (same learning parameters

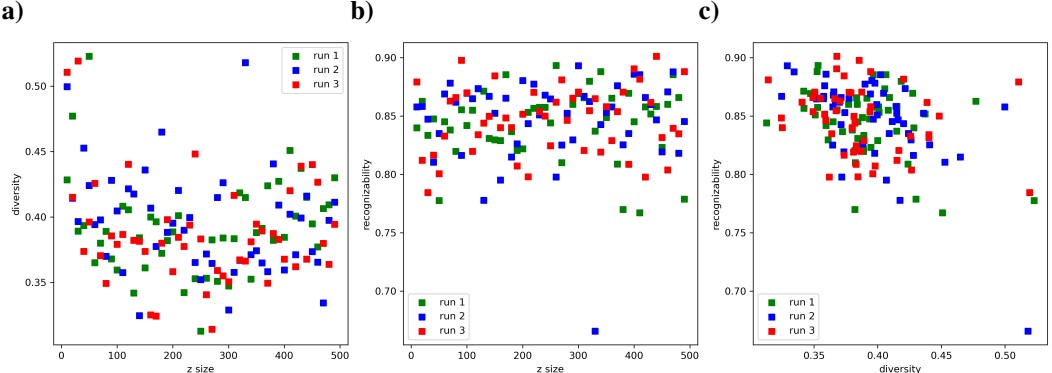

Figure S20: Effect of varying the size of the latent vector ($z$) in the **DA-GAN-RN** on the diversity/recognizability framework for 3 different runs. (**a**) Effect of latent size on the diversity. (**b**) Effect of the latent size on the recognizability. (**c**) Parametric curve recognizability versus diversity when one varies $\beta$ from 10 from to 500.

than those described in Section S1). Train/test loss and classification accuracy are reported for all training epochs in Fig S21a and Fig S21d, respectively.

- **A one-shot classifier**. Both the architecture and the training procedure of this classifier are described in Section S1. We remind the reader that we use a weak generalization split to train the few-learning networks (i.e. $1473$ classes in the training set and $150$ classes of testing set). Train/test loss and classification accuracy are reported for all training epochs in Fig S21b and Fig S21e, respectively.

- **A five-shot classifier**. This network is the exact same than the one-shot Prototypical Net described before, except that it is trained in a 5-shots settings. Train/test loss and classification accuracy are reported for all training epochs in Fig S21c and Fig S21f, respectively.

For the standard classifier, we observe an increase of the test loss (resp. a decrease of the test accuracy) while the train loss is still decreasing (resp. the train accuracy is still increasing), see Fig S21a and Fig S21d. It suggests that the network becomes better at classifying the training samples but worst at dealing with the testing samples. The standard classifier is then overfitting on the training set. Note that the 2 other few-shots learning networks are not showing such a decrease in the test loss and accuracy. Such an experiment suggests that standard classifiers are not adequate to extract features of samples in a low-data regime.

## S18  Computational Resources

All the experiments of this paper have been performed using Tesla V100 with 16 Gb memory. The training time is dependent on the hyper-parameters, but varies between 4h to 24h per simulation.

## S19  Broader Impact

This work does not present any foreseeable negative societal consequences. We think the societal impact of this work is positive. It might help the neuroscience community to evaluate the different mechanisms that allow human-level generalization, and then better understand the brain.

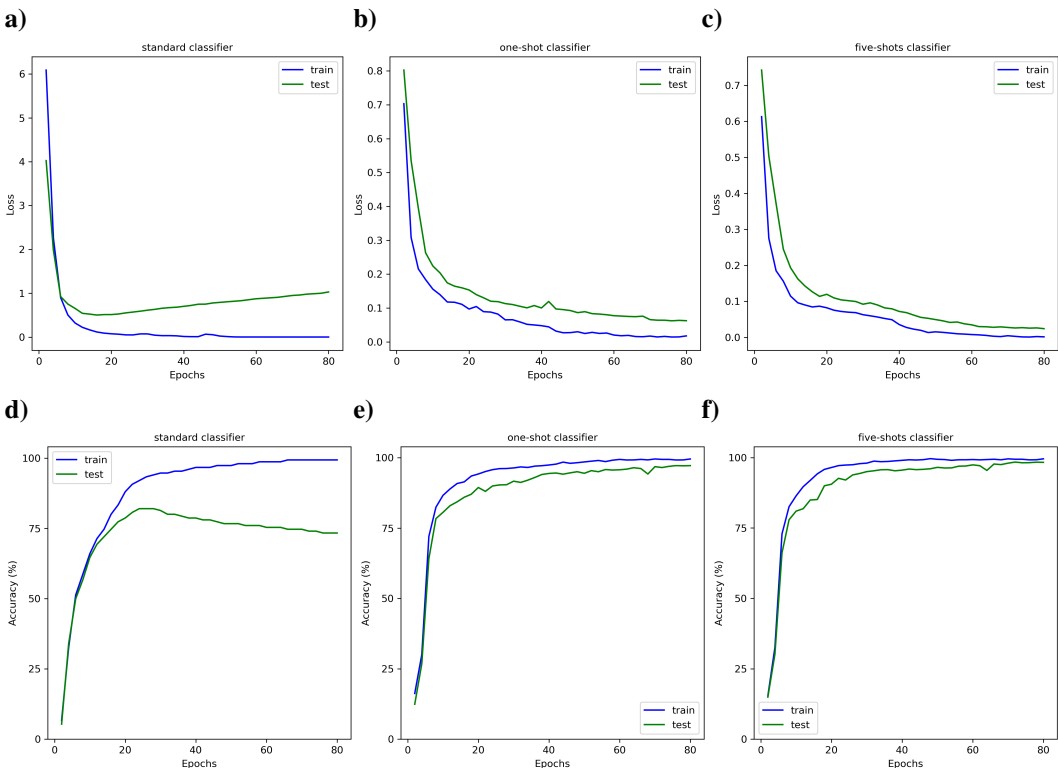

Figure S21: Comparison between different classifiers in low-data regime. Train and test losses at each training epoch for (**a**) a standard classifier, (**b**) a Protypical Net in a one-shot learning setting and (**c**) a Prototypical Net in a 5-shots learning setting. Train and test classification accuracy at each training epoch for (**d**) a standard classifier, (**e**) a Protypical Net in a one-shot learning setting and (**f**) a Prototypical Net in a 5-shots learning setting.