# OpenReview forum: "Diversity vs. Recognizability: Human-like generalization in one-shot generative models"
_NeurIPS.cc/2022/Conference — NeurIPS 2022 Accept_

### Official Review · Reviewer_fqWo · 2022-06-29

**Rating:** 7
**Confidence:** 4
**Soundness:** 4 excellent
**Presentation:** 3 good
**Contribution:** 3 good

**Summary:**

In this paper, the authors propose a new framework for assessing the quality of one-shot generative models for images. Particularly, the authors frame it as a two-object metric where diversity (how generated examples differ from each other) and accuracy (whether the generated example resembles the given single example) of the generative model are measured. Furthermore the human-made examples can also be put into comparison. The authors conduct the experiments of applying such two-object metric to representative one-shot generative models with detailed analysis and discussion.

**Questions:**

- Why using a one-shot classification model (MAML/Prototypical Net) to measure the Recognizability (mentioned in paragraph starting on line 204), instead of a full-fledged classifier trained on the whole dataset (e.g. including all data points of the  class that the one-shot example belongs to)? After all, the latter should have a better quality, and there seems to be nothing preventing the use of it, given that the proposed framework is a metric rather than a one-shot method.
- In the motivation (Sec 2.1) the author claims that likelihood-based algorithms or FID-like metrics are not ideal for the test. However this seems to be only argued but not supported by any experiments. Is there any minimal case that can demonstrate its failure in the proposed setting?
- Is the proposed method applied to other one-shot settings, such as one-shot image segmentation, etc?


**Limitations:**

No issue.

**Strengths And Weaknesses:**

Strengths of this work:
- It addresses the (relatively) less studied field of one-shot image generation and presents a framework as a metric for assessing the quality. The idea of using two objects is novel.
- This work is more of an empirical work, and it does a good job in clearing presenting experiments and the analysis. Comparison of representative one-shot image generation methods (with human performance comparable) are shown visually and the discussion is substantial. Overall, the writing quality is high and the content is substantial.
- The result presented in this paper would provide interesting insights in the one-shot image generation.


Weaknesses of this work (If they are properly addressed, I’m happy to increase my rating):
- Lack of comparison with previously leveraged metrics. It’s true that the proposed method is more holistic than the scattered ones used in previous works for evaluation, such as T-SNE, clustering, however readers would be also interested in how the new framework compass with such measures, for example how diversity and/or accuracy collrates with clustering results.
- The dataset is limited to Omniglot. Other datasets used in one-shot image generation (for example, CIFAR) should also be  considered as they may show different behaviors, which is the focus of what a metric would be studying. In such a case, human output is not feasible and thus not in the comparison, but still the impact of different methods, especially with different hyper parameters where the paper paid a lot of attention, would have significance.
- Some design choices are not clear (see Questions below)


# Changes during/after rebuttal:
Increased score from 6 to 7.

---

> ### Author Response · Authors · 2022-07-31
> **Response to reviewer fqWo**
>
> We thank the reviewer fqWo for the detailed review and the relevant comments. Please find below our answer :
>
> * Reviewer: “Other datasets used in one-shot image generation (for example, CIFAR) should also be considered as they may show different behaviors, which is the focus of what a metric would be studying”.
> A similar comment has been raised by reviewer D3RB. For readability we copy-paste the answer made to reviewer D3RB :
> The goal of this article is to introduce and to test a new framework that serves as a basis for the analysis for one-shot generative models. To test and robustify our framework, we have run 8 different control experiments (with different feature extractors, different diversity metrics, different type of augmentation for the contrastive loss, and different few-shot classifiers for the recognizability metric), we have done the systematic comparison of more than 900 models corresponding to approximately 5000 hours of compute. The result is an article with 43 graphs, 6 figures showcasing generated samples, 25 pages of Supplementary information… Therefore, our results are strong, and detailed. They demonstrate clearly the usefulness of our framework on a large variety of algorithms trained on Omniglot. The point of this article is not to propose a new algorithm, but a framework to study existing ones. In that sense, adding other datasets really is orthogonal to our main contribution (as stated by reviewer 69WY).
>
> * Reviewer "In the motivation (Sec 2.1) the author claims that likelihood-based algorithms or FID-like metrics are not ideal for the test. However this seems to be only argued but not supported by any experiments."
> This question was also raised by reviewer D3RB. For the sake of readability we copy-paste our previous answer:
> FID/IS methods involve first extracting features and then computing a statistical distance. Both processes are biased in the case of the Omniglot data:
> First, the feature extraction process relies on a fully feedforward classifier (Inception Net) that is known to overfit in low-data regime (which is a problem for Omniglot with only 20 samples per class). To convince the reviewer that standard classifier are overfitted (even on the weak generalization setting), we have run a simple experiment: we have compared 3 networks with the exact same architecture, one trained as a classical feedforward classifier (as Inception Net), another as a one-shot classifier (Prototypical Net) and the last one as a 5-shots classifier. As shown in Fig S21, the overfitting is clear in the case of the standard classifier: the test loss is increasing while the training one is still decreasing (This phenomenon has been mentioned in numerous articles [1], [2]...). Note that both one-shot and few-shots classifiers are not suffering from such an overtting (see Fig S21).
> Second, the computation of the statistical distance (either KL distance for IS or Wassertein for FID) needs plenty of samples to be unbiased. This specific point has been thoroughly studied in [3]. In our case, we need to compute class specific FID, and we have access to only 20 samples per class which is not enough to have an unbiased estimator of the statistical distance.
> For these 2 reasons the FID/IS are not adequate for the few-shot datasets. We have updated the 2.1 section (line 103-104) and added details and results of these new  experiments in the supplementary information (section S17).
>
> * Reviewer: “Why using a one-shot classification model (MAML/Prototypical Net) to measure the Recognizability (mentioned in paragraph starting on line 204), instead of a full-fledged classifier trained on the whole dataset (e.g. including all data points of the class that the one-shot example belongs to)?”
> This question is related to the previous one. A fully-fledged classifier would tend to overfit on Omniglot as we are in a low-data regime. We have run an additional experiment to illustrate this point, please refer to section S17 for more details.
>
> * Reviewer: "Is the proposed method applied to other one-shot settings, such as one-shot image segmentation, etc?"
> No, it is not. For now the framework only applies to one-shot drawing.
>
>
> We thank the reviewer fqWo for the sharp comments that help us to improve and clarify our article. We hope we have properly addressed the concerns of the reviewer, and we would be happy to initiate a discussion if it is not the case.
>
> [1] : Jadon, Shruti. "An overview of deep learning architectures in few-shot learning domain." arXiv preprint arXiv:2008.06365(2020).
> [2] : Brigato, Lorenzo, and Luca Iocchi. "A close look at deep learning with small data." 2020 25th International Conference on Pattern Recognition (ICPR). IEEE, 2021.
> [3] : Lake, Brenden M., Ruslan Salakhutdinov, and Joshua B. Tenenbaum. "Human-level concept learning through probabilistic program induction." Science 350.6266 (2015): 1332-1338.

---

> > ### Comment · Reviewer_fqWo · 2022-08-07
> > **Response to the rebuttal**
> >
> > I would like to thank the authors for the rebuttal that is very substantial (great job), and with a good amount of effort in making a clarification. I would encourage the author to concisely articulate the such clarification in a revised version of the paper, in which case the work would be strengthened.
> >
> > My concerns are addressed --- Increased my score from 6 to 7.

---

> > > ### Author Response · Authors · 2022-08-08
> > > **Response to reviewer fqWo**
> > >
> > > We thank the reviewer fqWo for increasing his/her score. If the paper is accepted, we will definitely use the additional 1-page to include more detailed clarification.

---

### Official Review · Reviewer_D3RB · 2022-07-04

**Rating:** 6
**Confidence:** 4
**Soundness:** 3 good
**Presentation:** 3 good
**Contribution:** 3 good

**Summary:**

This paper studies the problem of few-shot generation on the Omniglot dataset, with a focus of benchmarking existing algorithms against humans. They claim that existing metrics for generative models aren’t appropriate for evaluating one-shot generalization, and propose a different framework for this purpose. Their framework is based on two aspects: diversity and recognizability. In both cases, measuring these aspects is dependent upon a trained model. In the case of diversity, a feature extractor is used to measure intra-class variability (standard deviation of produced samples in the feature space around a ‘prototype’), and a SimCLR encoder is used for this. For recognizability, they make use of a one-shot classification model, and the accuracy of that model on generated samples is the recognizability measure. They use a Prototypical Network for this. They investigate where VAE-based and GAN-based models fall in this evaluation framework, and how this is affected by various hyperparameters. Due to the setup of the Omniglot dataset, it is also possible to compare against humans (Omniglot contains human-drawn characters which can be thought of as being the result of a human generative process). They found that the neural statistician-based VAE is the closest (on average) to humans in the diversity-recognizability space.

**Questions:**

It’s an interesting finding that VAE-STN and VAE-SN better approximate humans for different subsets of visual classes. Would be interesting to see a similar experiment including GAN models too. Also, the authors hypothesize that this is due to the attention of STN providing a better human approximation for simple visual concepts. Isn’t this counterintuitive, as attention can help to break down a complicated image by sequentially looking at different parts? It would be nice to run some additional experiments to examine this hypothesis, e.g. keeping the architecture the same as VAE-STN but removing the attention mechanism and seeing the effects of that on which types of visual contexts it has smallest distances to humans after this modification. It would also be nice to investigate this as a function of the number of attention steps.

Isn’t it counterintuitive that increasing the context size reduces the diversity, since allowing the model to see more examples should increase its understanding of intra-class variability?

Is there intuition on why varying the number of attentional steps for VAE-STN leads to a parabola in the recognizability/diversity space? What happens at the turn-around point? What is the significance of this finding?

How come different architectures are used for GAN models but not for VAE models?


# after the rebuttal
increased my score to a weak accept based on the discussion with the authors (please refer to my response to the authors for justification)

**Limitations:**

Aside from the limitations mentioned earlier in the review, another one is the following.
What if instead of a single image, a few images were available of the new visual concept (few-shot learning instead of one-shot learning)? The proposed framework assumes that in that case they will have to be aggregated into a prototype, but this is not necessarily the best approach. It would be nice to make it more flexible to allow evaluating any method for exploiting more than one ‘shots’.


**Strengths And Weaknesses:**

# Strengths:
[+] The paper is clearly written for the most part (see a couple exceptions in the detailed comments below) \
[+] Drawing connections between humans and different generative models is interesting  \
[+] The proposed framework is well-grounded, in that it draws inspiration from psychophysics work

# Weaknesses:
[-] Limited scope \
[-] Poorly motivated \
[-] Limited significance in the current form \


Detailed comments:
================

Clarity
=====
In Figure 4, it would be clearer to actually show the size of the context set, e.g. via the size of the circle, or color or something else. Currently I assume the direction of the arrow is giving a proxy for this but this is not as informative.

In Figure 5, in the caption: ‘Bigger squares correspond to the model showing the smallest distance to humans’. This is unclear as it sounds like the bigger the square, the smaller the distance. But these squares for the different models all have the same size and instead it’s their (y-axis) position that captures the distance. Should rephrase to ‘the squares correspond to…’ (i.e. remove the word bigger), if my understanding is correct.

As a high-level comment, it would have been useful to give some more details about the models used. For example, it is hard to understand exactly what effect we expect the context size to have on VAE-NS without a good enough understanding of that model.


Motivation and significance
=====================
To motivate their evaluation framework, the authors claim that existing metrics don’t work well if the training and testing samples are too dissimilar. In Omniglot though, this isn’t really the case. There have been several studies that show that in fact, feature reuse is sufficient for good few-shot learning performance on simple datasets like Omniglot (see [1] for an example in the context of MAML, [2] for a more general example), and in fact it was also demonstrated that good performance can even be achieved on new Omniglot classes without using the support labels at test time [3], which also hints at the simplicity of this generalization challenge. So, is this claim just a hypothesis, or is it based on some evidence on the failure of existing metrics for one-shot generation evaluation on Omniglot? Especially in the weak generalization setting considered here, this might not be an issue.

Further, for measuring recognizability, it wasn’t clear to me why we need to use a one-shot model as the ‘critic’ for whether the generated samples can be accurately classified. Why can’t we use an ‘oracle’ here that was trained on more shots from the new classes? One might argue that this is okay since this model is used just for evaluation purposes (though perhaps we do want an *example-level* split where e.g. this oracle is trained on support images but not query images from the new classes). The current setup seemed unnecessarily restrictive to me and not well motivated. If we can use an oracle trained on more shots, this also sidesteps the other difficulty mentioned by the authors as a limitation that prevents us from using existing metrics / evaluation framework for this problem.

Finally, why did the authors decide to use a different model (SimCLR) to get the encoder used for measuring diversity compared to the model used for recognizability (Prototypical Networks)? Can’t we use the same model for both of these? (and ideally report results with different choices for this model)

Limited scope
===========
Despite this paper being largely empirical, there is a limited number of models experimented with. Why not also compare against the few-shot generalization model in [4], for example?

Omniglot is also a very simple dataset and few-shot learning research (at least for classification) has drifted from it these days [5]. It would be nice to report results on more modern datasets. Quickdraw, which is mentioned in the discussions, can be a good option as it still allows comparisons to humans.

Significance
===========

All in all, due to the limited scope and insufficient motivation of the proposed framework and set of experiments, I felt that the significance of the findings is low. Further, some findings like GANs lacking diversity are already known (e.g. mode collapse is a known problem for GANs). Verifying that the same is true in the one-shot scenario is interesting, but not too surprising, especially in this weak generalization case studied in this paper.

References
=========
[1] Rapid Learning or Feature Reuse? Towards Understanding the Effectiveness of MAML. Raghu et al. ICLR 2020.
[2] A Closer Look at Few-shot Classification. Chen et al. ICLR 2020.
[3] Are Few-Shot Learning Benchmarks too Simple ? Solving them without Task Supervision at Test-Time. Huang et al.
[4] One-shot Generalization in Deep Generative Models. Rezende et al. ICML 2016.
[5] Meta-Dataset: A Dataset of Datasets for Learning to Learn from Few Examples. Triantafillou et al. ICLR 2020.

---

> ### Author Response · Authors · 2022-07-31
> **Response to reviewer D3RB (1/4)**
>
> We thank the reviewer D3RB for the very detailed review as well as the constructive suggestions to improve the paper. Please find below our point by point answer that follows the same order as the reviewer’s comments :
>
> ### Clarity:
> We have modified our article to include all the reviewer’ comment to improve clarity. More specifically:
> + Reviewer: “Currently I assume the direction of the arrow in Fig 4 is giving a proxy for this but this is not as informative.”
> We have tried to apply the reviewer’s idea to make the diameters of the points proportional to the context size in Fig 4-a. But the resulting plot is actually confusing because all the data points corresponding to high context size are located at the same location. We have added a sentence in the caption to clarify the meaning of the arrows in both Fig 4.
>
> + Reviewer : “In Figure 5, the caption: ‘Bigger squares correspond to the model showing the smallest distance to humans’ is unclear as it sounds like the bigger the square, the smaller the distance.”
> The reviewer is right and we have updated the caption to improve clarity.
>
> + Reviewer : “As a high-level comment, it would have been useful to give some more details about the models used”:
> Note that because of the limited space, all the model fine details were pushed to the supplementary information. To facilitate a better understanding of the models we have further added figures in the supplementary materials (see section S9 and S10)  -- to provide a quick overview of the models and highlight the specificity of each model. Concerning the context inclusion mechanism, we have added a small section at line 738.
>
> We sincerely thank the reviewer for these suggestions of improvement.

---

> > ### Author Response · Authors · 2022-07-31
> > **Response to reviewer D3RB (2/4)**
> >
> > ### Motivation and significance
> > * Reviewer: The authors should better motivate the fact that existing metrics are not fit for one-shot generation on Omniglot dataset (especially in weak generalization settings).
> > FID/IS methods involve first extracting features and then computing a statistical distance. Both processes are biased in the case of the Omniglot data:
> > First, the feature extraction process relies on a fully feedforward classifier (Inception Net) that is known to overfit in low-data regime (which is a problem for Omniglot with only 20 samples per class). To convince the reviewer that standard classifier are overfitted (even on the weak generalization setting), we have run a simple experiment: we have compared 3 networks with the exact same architecture, one trained as a classical feedforward classifier (as Inception Net), another as a one-shot classifier (Prototypical Net) and the last one as a 5-shots classifier. As shown in Fig S21, the overfitting is clear in the case of the standard classifier: the test loss is increasing while the training one is still decreasing (This phenomenon has been mentioned in numerous articles [1], [2]...). Note that both one-shot and few-shots classifiers are not suffering from such an overtting (see Fig S21).
> > Second, the computation of the statistical distance (either KL distance for IS or Wassertein for FID) needs plenty of samples to be unbiased. This specific point has been thoroughly studied in [3]. In our case, we need to compute class specific FID, and we have access to only 20 samples per class which is not enough to have an unbiased estimator of the statistical distance.
> > For these 2 reasons the FID/IS are not adequate for the few-shot datasets. We have updated the 2.1 section (line 103-104) and added details and results of these new  experiments in the supplementary information (section S17).
> >
> > * Reviewers: “Further, for measuring recognizability, it wasn’t clear to me why we need to use a one-shot model as the ‘critic’ for whether the generated samples can be accurately classified”.
> > The reviewer is right that the one-shot model critic could easily be replaced by a few-shot model critic. The reason behind our choice was to match the setting proposed by [4]. We have updated the article to better explain this (see line 206-208). Note that a classical feedforward oracle classifier (which is not trained for one-shot or few-shot classification) would be inappropriate in our case because such networks are known to overfit to the training data in low-data regime (Omniglot has only 20 samples per category). But both one-shot and few-shot classifiers are not showing any overfitting sign (see Fig S21). We have expanded the main text and included these additional experiments in the supplementary material (in section S17).
> >
> > * Reviewer: “Finally, why did the authors decide to use a different model (SimCLR) to get the encoder used for measuring diversity compared to the model used for recognizability (Prototypical Networks)?”.
> > We did not use ONLY SimCLR for diversity evaluation. As shown in the Supplementary information (see section 3.1) and explained in the main text (line 194), we have considered both SimCLR and Prototypical Net as encoders to compute the diversity. We have observed a strong correlation between the diversity computed by both encoders (correlation of 0.86), meaning that using one or the other encoder does not change the diversity measure. We have decided to report the diversity measure as computed by the SimCLR network in the main article because it relies on a fully unsupervised method (contrastive loss), and it is, in our opinion, a more appealing feature extraction method than the one derived from the Prototypical Net (which needs supervision labels). We agree with the reviewer that Prototypical Net could be used to assess both accuracy and diversity (that is why we have run such extensive control experiments in supplementary materials).
> >
> > [1] : Jadon, Shruti. "An overview of deep learning architectures in few-shot learning domain." arXiv preprint arXiv:2008.06365(2020).
> > [2] : Brigato, Lorenzo, and Luca Iocchi. "A close look at deep learning with small data." 2020 25th International Conference on Pattern Recognition (ICPR). IEEE, 2021.
> > [3] : Chong, Min Jin, and David Forsyth. "Effectively unbiased fid and inception score and where to find them." Proceedings of the IEEE/CVF conference on computer vision and pattern recognition. 2020.

---

> > > ### Author Response · Authors · 2022-07-31
> > > **Response to reviewer D3RB (3/4)**
> > >
> > > ### Limited Scope
> > > * Reviewer: “Why not also compare against the few-shot generalization model in [6], for example?”
> > > This point is rather puzzling as we HAVE implemented the model proposed from [6] (see line 123). In the entire article, we refer to this model as the ‘VAE-STN’ (to keep a compact name). Note that as the authors in [6] did not name their model, we have used the ‘VAE-STN’ name.
> > >
> > > * Reviewer: “It would be nice to report results on more modern datasets. Quickdraw, which is mentioned in the discussions, can be a good option as it still allows comparisons to humans.”
> > > The goal of this article is to introduce and to test a new framework that serves as a basis for the analysis for one-shot generative models. To test and robustify our framework, we have run 8 different control experiments (with different feature extractors, different diversity metrics, different type of augmentation for the contrastive loss, and different few-shot classifiers for the recognizability metric), we have done the systematic comparison of more than 900 models corresponding to approximately 5000 hours of compute. The result is an article with 43 graphs, 6 figures showcasing generated samples, 25 pages of Supplementary information… Therefore, our results are strong, and detailed. They demonstrate clearly the usefulness of our framework on a large variety of algorithms trained on Omniglot. The point of this article is not to propose a new algorithm, but a framework to study existing ones. In that sense, adding other datasets really is orthogonal to our main contribution (as stated by reviewer 69WY).
> > >
> > > ### Significance
> > > * Reviewer: “Further, some findings like GANs lacking diversity are already known (e.g. mode collapse is a known problem for GANs). Verifying that the same is true in the one-shot scenario is interesting, but not too surprising, especially in this weak generalization case studied in this paper”.
> > > With all due respect, our manuscript goes well behind showing that “GANs lack diversity”  as we introduce a novel framework that can be used to compare and quantify, among other things,  lack of sample diversity and/or recognizability for different one-shot generation algorithms for a direct comparison with humans. To the best of our knowledge, this is the first time such a framework is proposed, and applied to compare SOTA VAE and GAN architectures (not only in terms of diversity, but also recognizability). Consequently, we do think our contribution is significant, and we hope the reviewer will reassess his evaluation of our work.
> > >
> > > [6] : Rezende, Danilo, et al. "One-shot generalization in deep generative models." International conference on machine learning. PMLR, 2016.

---

> > > > ### Author Response · Authors · 2022-07-31
> > > > **Response to Reviewer D3RB (4/4)**
> > > >
> > > > ### Questions:
> > > > * Reviewer: “Isn’t it counterintuitive that increasing the context size reduces the diversity, since allowing the model to see more examples should increase its understanding of intra-class variability?”
> > > > Our intuition to answer this interesting question is borrowed from  [5]. First, we need to keep in mind that in the VAE-NS, the context size is the number of examples used to define a given concept during training (not testing, for which we always have one single exemplar for the one-shot generation task). That being said, when the context size is one, it is virtually impossible to properly assess the intra-class variability of the training concepts. In such a case, it has been theoretically shown that a Bayesian observer will tend to overestimate the intra-class variance [5]. In [5], the authors have also shown that the more samples are used to define a concept (i.e. the higher the context size), the more the estimated variance decreases to ultimately reach the ‘true’ variance of the category. This is what we observe for the VAE-NS, suggesting that the VAE-NS is acting as a Bayesian observer ! We have extended the discussion to include this point (see line 333-336).
> > > >
> > > > * Reviewer: “Is there intuition on why varying the number of attentional steps for VAE-STN leads to a parabola in the recognizability/diversity space? What happens at the turn-around point? What is the significance of this finding?”
> > > > We think the parabola observed in the recognizability/diversity space is an artifact of the LSTM used in VAE-STN networks. In the VAE-STN each attentional step corresponds to a recurrence step in the LSTM. After a certain number of steps (here approximately 60-70 steps) the LSTM might not be able to capture long-term dependencies between the last and the first time steps.
> > > >
> > > > * Reviewer: “How come different architectures are used for GAN models but not for VAE models?”
> > > > We DID compare different architectures for the VAE models (in Fig. 3a only), but we failed to make it clear in the paper ! We have varied the size of the latent space of the static-network (h in section S9 of the Supplementary material) as well as the size of the latent space of the inference network (z in section S9 of the supplementary material) for the VAE-NS. In the case of the VAE-STN we have varied the size of the latent space (sz in section S8.1 of the Supplementary Material). Note that we did not include a comparison of the different architectures in the diversity/recognizability space because the effect seems to be limited (see section S16). We apologize for this lack of clarity, and we have updated the main article (line 293-295) to make this point clearer. Note also that in the case of the GANs-like models we have presented 2 different backbones (U-Net and ResNet) because it was presented similarly in the DAGAN article [7].
> > > >
> > > > We think our comments have addressed most of the concerns raised by the reviewer D3RB. If not, we hope the reviewer D3RB is going to initiate a discussion so that we can answer all the remaining questions and convince him to increase his rating. We thank the reviewer D3RB for helping us to push our work towards better clarity.
> > > >
> > > >
> > > > [5] : Tenenbaum, Joshua. "Bayesian modeling of human concept learning." Advances in neural information processing systems11 (1998).
> > > > [7] : Antoniou, Antreas, Amos Storkey, and Harrison Edwards. "Data augmentation generative adversarial networks." arXiv preprint arXiv:1711.04340 (2017).

---

> > > > > ### Comment · Reviewer_D3RB · 2022-08-09
> > > > > **thank you for the thorough responses!**
> > > > >
> > > > > I'd like to thank the authors for the thorough responses, clarifications and additional experiments. They have convinced me to increase my score to a weak accept, especially with the additional experiments that clearly show the overfitting effect preventing traditional metrics to be used for evaluation (I feel that this is a very important point and addition to the paper as it nicely motivates the proposed framework), the clarification that VAE-STN refers to the model from [6] (I had missed that!) and responses to most of the points I raised. Below are some additional thoughts.
> > > > >
> > > > > I still do feel that the scope is limited. While the authors have undoubtedly ran a very large number of experiments, these are performed in a very simple dataset, and it is definitely the case that the generalization problem on Omniglot is a weak one (test classes don't look too dissimilar to training classes), as has been argued in several few-shot learning papers in the context of classification (e.g cited in my original review). However, it is also true that *generation* is significantly less well-studied than classification (and presumably significantly harder both to create and to evaluate models), so I do appreciate that the simpler benchmark is justified to some extent here. Though I would really like to see future work tackling harder problems.
> > > > >
> > > > > RE: oracles vs one-/few- shot learning models. I understand that to compute traditional metrics, a feature extraction process is used which relies on an Inception Net, that may overfit in the low-data regime. However, it seems to me that there is a simple workaround here: we can evaluate one-shot generalization models on datasets that do have large number of examples per classes, but only one or a few shots are shown to the model whose one-shot generalization performance we want to evaluate. That is, instead of using datasets that are truly few-shot (on Omniglot, there are only 20 examples per class available), we can use larger ones, allowing to train the required feature extractor for evaluation without overfitting, but only show the model limited data per class, to make it one-/few- shot from its perspective. Does this setup make sense, and would it address the issues with traditional evaluation metrics? Would be great to discuss other evaluation setup alternatives more. However, I do see that the proposed framework is interesting in that it's grounded in phychophysics, and has the advantage of facilitating comparisons with human generalization, so I do think there is merit to it regardless.

---

> > > > > > ### Author Response · Authors · 2022-08-09
> > > > > > **Thanks for the response**
> > > > > >
> > > > > > We thank the reviewer D3RB for the response as well as the increase of the score.
> > > > > >
> > > > > > We agree with the reviewer that the additional experiment we have run during the rebuttal is important to better motivate the article ! Thus, If the paper is accepted, we plan to use the additional 1-page to detail and include the results of this additional experiment.
> > > > > >
> > > > > > We also agree that testing our framework on other datasets is the next-step to go. We are planning to write follow up paper with more datasets and more models.
> > > > > >
> > > > > > The point raised by the reviewer D3RB concerning the alternative to the proposed evaluation metric is interesting. In our case this is hardly applicable (because of the monoglot dataset). But as we plan to retrain our models on QuickDraw for follow-up work we will be able to test and compare the reviewer's D3RB alternative with our current evaluation method. We plan to include this comparison in our future work. Thank you for this great idea !
> > > > > >
> > > > > > We have appreciated the very detailed comments of the reviewer, and we think the reviewer D3RB has helped us to greatly improve our work. Thank you !

---

### Official Review · Reviewer_69WY · 2022-07-08

**Rating:** 8
**Confidence:** 4
**Soundness:** 4 excellent
**Presentation:** 4 excellent
**Contribution:** 4 excellent

**Summary:**

There exist metrics for measuring the quality and consistency of standard generative models, but these are not applicable to the one-shot setting. Therefore, the authors of this paper introduce and quantify a diversity-recognizability space where single shot generated samples can be evaluated. Furthermore, these two axes make it straight-forward to compare the generated samples to samples from a human model (a brain). Extensive experiments establish the solidity of these measures for the Omniglot dataset.

**Questions:**

1.It is mentioned on lines 272-274 that the number of attentional steps is less imporatnt than the number of context samples; how do you see the relevance of $\beta$ relative to the number of context samples?

2. How relevant is context for the Omniglot dataset? Visibly, it does play a role in the experiments, but I am curious how you interpret it.

3. In the last paragraph of the conclusions, you place the most importance on creativity and the diversity part of your introduced metric. Does this mean that you think that the field should move away from GAN-based models when it comes to human-like generated data?


**Limitations:**

Yes

**Strengths And Weaknesses:**

This is a strikingly solid paper, which presents a clear and novel idea. The result on how GAN-based architectures and VAE-based approaches place in different parts of the diversity-recognizability scale is highly interesting. Each experiment is well-motivated (Section 3), and their results are thoroughly transparent and well discussed. The diversity-recognizability space also presents a very appealing way to think about evaluating generative models, well situated in the psychophysics and cognitive science literature, to the best of my knowledge. Relevant architectures and methods are used both for the feature extraction as part of the metrics, as well as for the one-shot generation. It is difficult to find something to remark on. The one thing would be the fact that the method is run on a single dataset, but in my view this should be left for future work, since a very clear and useful framework is introduced here for anyone to pick up – and the experiments are extensive even for this one dataset. Thus, I do not count this as a weakness. Last, the paper is highly readable and well-structured. I believe that it can be of large value to the community.

---

> ### Author Response · Authors · 2022-07-31
> **Response to reviewer 69WY**
>
> We thank the reviewer 69WY for such an encouraging and positive review ! Please find below the answer to your questions:
>
> * Reviewer : “How do you see the relevance of β relative to the number of context samples?”.
> In Fig 4-a we can observe (for the VAE-NS) that an increase in the number of context samples causes a drastic increase in recognizability (from 50% to 96%) and an important decrease of the Diversity (from 2.4 to 1.2). In Fig 4-b (still for the VAE-NS), varying β causes a change in recognizability that has a lower amplitude (from 77% to 92%) and a decrease of diversity (from 2.1 to 1.4). So it seems that the number of context samples has more impact on the Recognizability/Diversity  than β. We have added a sentence on line 290-292. Thanks for bringing up this interesting point.
>
> * Reviewer : “How relevant is context for the Omniglot dataset? Visibly, it does play a role in the experiments, but I am curious how you interpret it.”
> The context, in the frame of the VAE-NS, corresponds to the samples used during training, to evaluate the statistics of a specific category (i.e. a concept). In practice, we pass to the network different samples representing the same concept, and we vary the number of these samples (from 2 to 20; section 4.2). Intuitively, with more context samples for a given category, it becomes easier for the network to identify the properties and features that are crucial to define a given handwritten letter (which results in a higher recognizability but leaves less room for diversity). We have added a small section (see line 738) in the supplementary materials to clarify this point.
>
> * Reviewer: “Does this mean that you think that the field should move away from GAN-based models when it comes to human-like generated data?”
> If the point is to model the human generative process, then GANs-like models might not be a good choice as they do not account for the diversity level observed in humans. We do think that the ability of the brain to generate diverse and creative samples is crucial to explain such a good generalization ! We have added a sentence on line 343-346 to include this point.

---

> > ### Comment · Reviewer_69WY · 2022-08-09
> > **Thank you**
> >
> > Thank you very much to the authors for your replies to my questions! My positive rating is unchanged.

---

### Author Response · Authors · 2022-07-31
**Update Version**

Dear reviewers,

We have uploaded a track-changes version of the updated manuscript so that the reviewers can easily see the modifications we have made. In this version, sentences marked in blue have been added, and those in red have been removed (compared to previous version). The supplementary materials are appended at the end of the file.

---

### Meta-Review · Area_Chair_wBZq · 2022-09-01

**Recommendation:** Accept
**Confidence:** Certain

**Metareview:**

This paper studies the problem of few-shot generation on the Omniglot dataset, contrasting few-shot generative models against humans. They introduce "diversity" and "recognizability" metrics and perform an empirical analysis of how various models are situated in the diversity-recognizability plane relative to humans.

Overall this is a well-written paper, with a clear story and experiments. It provides interesting insights on how various generative models architectures relate to humans in a particular task.

**Award:**

No

---

### Decision · Program_Chairs · 2022-09-14

Accept